# Spectral index-flux relation for investigating the origins of steep decay in $\gamma$-ray bursts

Samuele Ronchini [1,2,3 ✉], Gor Oganesyan [1,2,3], Marica Branchesi[1,2,3], Stefano Ascenzi[4,5,6],
Maria Grazia Bernardini [4], Francesco Brighenti [1], Simone Dall'Osso [1,2], Paolo D'Avanzo[4],
Giancarlo Ghirlanda [4,7], Gabriele Ghisellini[4], Maria Edvige Ravasio [4,7] & Om Sharan Salafia[4,8]

$\gamma$-ray bursts (GRBs) are short-lived transients releasing a large amount of energy ($10^{51} - 10^{53}$ erg) in the keV-MeV energy range. GRBs are thought to originate from internal dissipation of the energy carried by ultra-relativistic jets launched by the remnant of a massive star's death or a compact binary coalescence. While thousands of GRBs have been observed over the last thirty years, we still have an incomplete understanding of where and how the radiation is generated in the jet. Here we show a relation between the spectral index and the flux found by investigating the X-ray tails of bright GRB pulses via time-resolved spectral analysis. This relation is incompatible with the long standing scenario which invokes the delayed arrival of photons from high-latitude parts of the jet. While the alternative scenarios cannot be firmly excluded, the adiabatic cooling of the emitting particles is the most plausible explanation for the discovered relation, suggesting a proton-synchrotron origin of the GRB emission.

[1] Gran Sasso Science Institute, L'Aquila, Italy. [2] INFN—Laboratori Nazionali del Gran Sasso, L'Aquila, Italy. [3] INAF—Osservatorio Astronomico d'Abruzzo, Teramo, Italy. [4] INAF—Osservatorio Astronomico di Brera, Merate, Italy. [5] Institute of Space Sciences (ICE, CSIC), Campus UAB, Barcelona, Spain. [6] Institut d'Estudis Espacials de Catalunya (IEEC), Barcelona, Spain. [7] Università degli Studi di Milano-Bicocca, Dip. di Fisica "G. Occhialini", Milano, Italy. [8] INFN-Sezione di Milano-Bicocca, Milano, Italy. ✉email: samuele.ronchini@gssi.it

The prompt emission of GRBs is characterized by an erratic superposition of several pulses, whose spectrum typically peaks in the keV–MeV energies. Its physical origin is still matter of discussion and the main open questions concern the composition of the jet (matter-[1] or magnetic[2]-dominated), the energy dissipation mechanisms (sub-photospheric emission[3], internal shocks[4] or magnetic reconnection[5]), and the nature of particle radiation. Once the prompt emission ceases, the light curve usually presents a steep decay (SD) phase[6–9] (tail), which can be well monitored in the X-ray band. The duration of the SD is around $10^2$–$10^3$ s and it is characterized by a typical decay power-law slope of 3–5. After the prompt emission, the jet interacts with the interstellar medium, producing the so called afterglow emission[10–12]. The afterglow models cannot account for such steep slopes and the origin of the SD is attributed to the fade-off of the emission mechanism that is responsible of the prompt phase.

Considering that the emitting surface of the jet is curved, an on-axis observer first receives photons from the line of sight and later photons from higher latitudes[13–15], which are less Doppler boosted. This gives rise to the so called high-latitude emission (HLE). Under the assumption of a single power-law spectrum ($F_\nu \propto \nu^{-\beta}$), the HLE predicts that the flux decays as $F_\nu(t_{obs}) \propto \nu^{-\beta} t_{obs}^{-(\beta+2)}$. On the other hand, if the spectrum is curved, the HLE can also lead to the transition of the spectral peak across the observing band[16], causing a spectral evolution, as often observed in the soft X-rays[17,18].

In this work, we find a unique relation between the spectral index and the flux. Here, we systematically analyze the X-ray spectral evolution during the SD phase as motivated by fact that temporal and spectral evolution during the tail of prompt pulses can provide clues about emission and cooling processes in GRB jets. Given the same trend followed by all the GRBs of our sample, we search for a common process at the basis of the spectral relation. We find that the standard HLE model cannot account for the observed relation, implying that efficient cooling of particles is disfavored. We test several assumptions about the dominating cooling mechanisms and we find that the combined action of adiabatic cooling of particles and magnetic field decay robustly reproduces our data. We conclude discussing the implications for the physics of GRB jets, their composition, and radiation mechanisms.

## Results

In order to investigate the spectral evolution during the SD phase, we select a sample of GRBs from the archive of the X-ray Telescope (XRT, 0.3–10 keV) on-board the Neil Gehrels Swift Observatory (Swift)[19]. We restrict our study to a sample of GRBs (eight in total) whose brightest pulse in the Burst Alert Telescope (BAT, 15–350 keV) corresponds to the XRT peak preceding the X-ray tail (see as example the Fig. 1a). We perform a time-resolved spectral analysis of the tail in the E = (0.5–10) keV band assuming a simple power-law model for the photon spectrum $N_\gamma \propto E^{-\alpha}$ (see "Methods:" "Sample selection," "Time-resolved spectral analysis," and "Spectral modeling"). We represent the spectral evolution plotting the photon index $\alpha$ as a function of the flux $F$ integrated in the E = (0.5–10) keV band, hereafter referred to as the $\alpha - F$ relation. The flux is normalized to the peak value of the X-ray tail. This normalization makes the result independent of the intrinsic brightness of the pulse and of the distance of the GRB.

We find a unique $\alpha - F$ relation for the analyzed GRBs as shown in Fig. 1b. This is consistent with a systematic softening of the spectrum; the photon index evolves from a value of $\alpha \sim 0.5$–1 at the peak of the XRT pulse to $\alpha \sim 2$–2.5 at the end of the tail

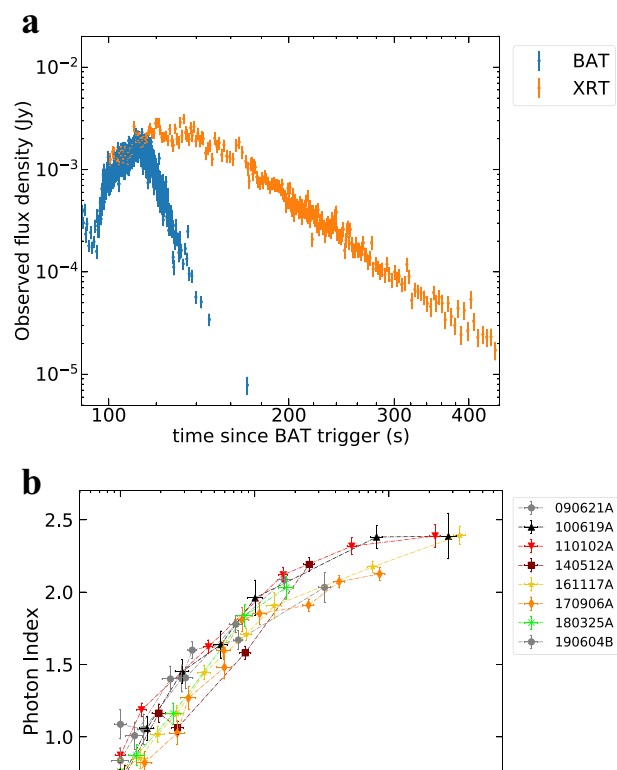

**Fig. 1 The steep decay phase and the correspondent spectral evolution.** In **a** we show an example of a light curve of an X-ray tail selected from our sample, taken from the GRB 161117A. We show on the same plot the XRT (orange) and the BAT (blue) flux density at 1 and 50 keV, respectively. The XRT light curve decays less steeply than BAT because of the evolution of the peak energy. The error bars represent 1σ uncertainties and they are derived from the Swift archive. In **b** we report the spectral evolution of the X-ray tail for all the GRBs in the first sample (shown with different colors). The photon index $\alpha$ is represented as a function of the reciprocal of the normalized flux $F_{max}/F$. Time flows from left to right. The error bars represent 1σ uncertainties, calculated via spectral fitting in XSPEC. In the legend, we report the name of each GRB.

emission, while the flux drops by two orders of magnitude. The initial and final photon indices are consistent with the typical low- and high-energy values found from the analysis of the prompt emission spectrum of GRBs, namely ~1 and ~2.3[20–22], respectively. The $\alpha - F$ relation can be interpreted as being due to a spectral evolution in which the spectral shape does not vary in time, but the whole spectrum is gradually shifted toward lower energies while becoming progressively dimmer (see Fig. 2). The consistent spectral evolution discovered in our analysis is a clear indication of a common physical mechanism responsible for the tail emission of GRBs and the corresponding spectral softening.

**Testing HLE**. We first compare our results with the expectations from the HLE, which is the widely adopted model for interpreting the X-ray tails of GRBs. When the emission from a curved surface is switched off, an observer receives photons from increasing latitudes with respect to the line of sight. The higher the latitude, the lower the Doppler factor, resulting in a shift toward lower energies of the spectrum in the observer frame. Through an

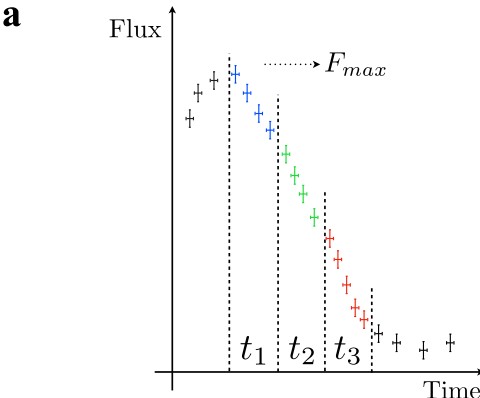

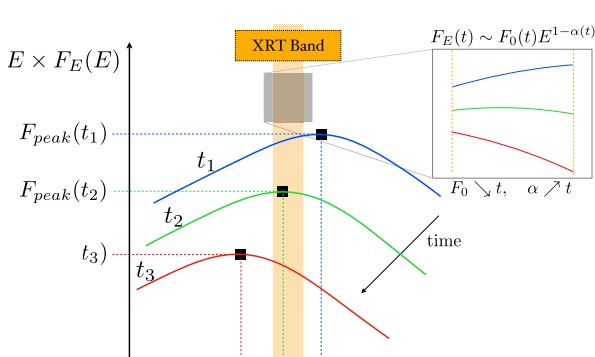

**Fig. 2 Illustration of the spectral evolution caused by a shift of the spectrum towards lower energies.** The transition of the spectral peak through the XRT band explains the observed spectral softening. The spectra in (**b**) colored in blue, green, and red correspond to the three temporal bins shown in (**a**) with the same colors. The inset in (**b**) shows how the local spectral slope evolves as observed in the XRT band. Since in the **b** we plot the flux density, the local slope in the XRT band is given by $1 - \alpha$, where $\alpha$ is the photon index. Both the x and y axes in (**a**) and (**b**) have arbitrary units.

accurate modeling of HLE (as described in "Methods:" "HLE from infinitesimal duration pulse") we derive the predicted $\alpha - F$ relation under the assumption of an abrupt shutdown of the emission, consistent with particles cooling on timescales much smaller than the dynamical timescale. We first consider a smoothly broken power-law (SBPL) comoving spectrum. Regardless of the choice of the peak energy, the bulk Lorentz factor or the radius of the emitting surface, the HLE predicts an $\alpha - F$ relation whose rise is shallower than the observed one (Fig. 3). We, additionally, test the Band function, commonly adopted for GRB spectra[23], and the physically motivated synchrotron spectrum[4], obtaining similar results (Supplementary Figs. 11 and 12a): the HLE softening is too slow to account for the observed $\alpha - F$ relation. We further relax the assumption of an infinitesimal duration pulse, i.e., considering a shell that is continuously emitting during its expansion and suddenly switches off at radius $R_0$[24] (see Supplementary Note 1). The contributions from regions $R < R_0$ are subdominant with respect to the emission coming from the last emitting surface at $R = R_0$, resulting in a spectral evolution still incompatible with the observations (Supplementary Fig. 1). An interesting alternative is the HLE emission from an accelerating region[25] taking place in some Poynting flux dissipation scenarios[5]. Even though it can explain the temporal

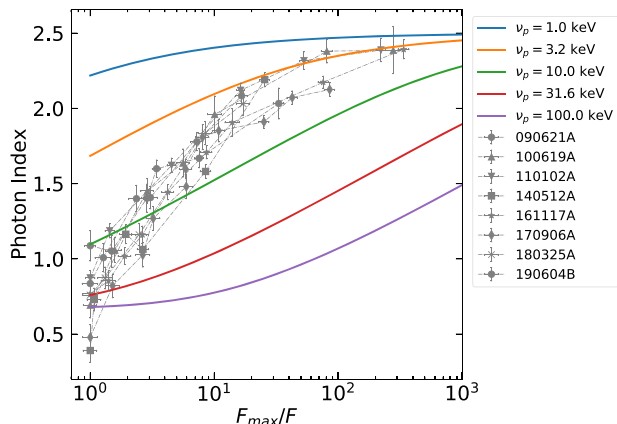

**Fig. 3 Spectral evolution expected for HLE from an infinitesimal duration pulse.** The comoving spectrum is assumed to be a SBPL. The several colors indicate the observed peak frequency at the beginning of the decay. The error bars represent $1\sigma$ uncertainties, calculated via spectral fitting in XSPEC. In the legend, we report the name of each GRB.

slopes observed in the X-ray tails, also this scenario fails in reproducing the $\alpha - F$ relation (see Supplementary Fig. 2). Our results on HLE are based on the assumption of a common comoving spectrum along the entire jet core. Even changing the curvature (or sharpness) of the spectrum or assuming a latitude dependence of the spectral shape, the disagreement with the data remains, unless we adopt a very fine-tuned structure of the spectrum along the jet core, which is not physically motivated (see "Methods:" "HLE from infinitesimal duration pulse"). Alternative models, such as anisotropic jet core[26–28] or sub-photospheric dissipation[29], can hardly reproduce our results (see Supplementary Note 3).

**Adiabatic cooling**. Since the standard HLE from efficiently cooled particles and its modified versions, as well as alternative scenarios, are not able to robustly interpret the observed $\alpha - F$ relation, we consider a mechanism based on an intrinsic evolution of the comoving spectrum. The most natural process is the adiabatic cooling of the emitting particles[30]. Here we assume conservation of the entropy of the emitting system $\langle\gamma\rangle^3 V'$ throughout its dynamical evolution, where $\langle\gamma\rangle$ is the average random Lorentz factor of the emitting particles and $V' \propto R^2\Delta R'$ the comoving volume[31]. We consider both thick and thin emitting regions, i.e., a comoving thickness of the emitting shell $\Delta R' = $ const or $\Delta R' \propto R$, respectively. We assume a power-law radial decay of the magnetic field $B = B_0(R/R_0)^{-\lambda}$, with $\lambda > 0$, and synchrotron radiation as the dominant emission mechanism. Here $R_0$ is the radius at which adiabatic cooling starts to dominate the evolution of the emitting particles. We compute the observed emission taking also into account the effect of HLE by integrating the comoving intensity along the equal arrival time surfaces (EATS) (see "Methods:" "Adiabatic cooling"). In this scenario, contrary to HLE alone, the emission from the jet is not switched off suddenly, but the drop in flux and the spectral evolution is produced by a gradual fading and softening of the source, driven by adiabatic cooling of particles. The resulting spectral evolution and light curves are shown in Fig. 4.

Adiabatic cooling produces a much faster softening of $\alpha$ as a function of the flux decay, with respect to HLE alone, in agreement with the data. Assuming a different evolution of the shell thickness, the behavior of the curves changes only marginally (see Supplementary Fig. 3). For large values of $\lambda$, the evolution of $\alpha$ flattens in the late part of the decay (see Fig. 4a),

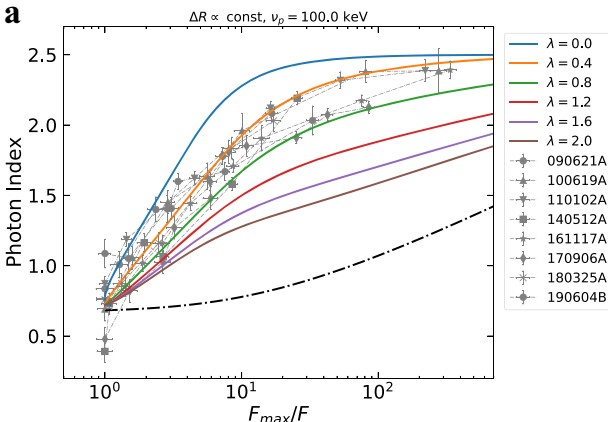

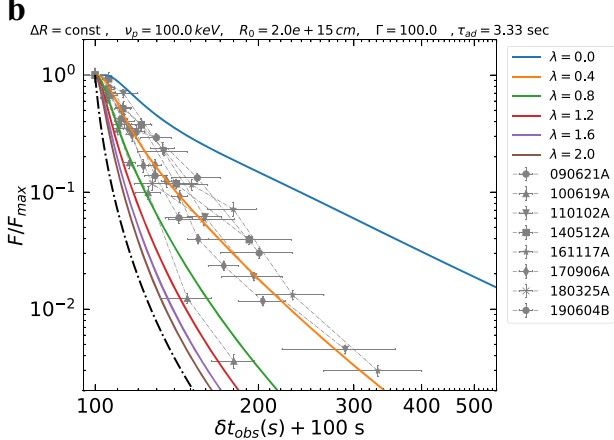

**Fig. 4 Spectral and temporal evolution in case of adiabatic cooling.** In **a** we show the $\alpha - F$ relation expected in the case of adiabatic cooling (solid lines). The theoretical curves are computed taking also into account the effect of HLE. The value of $\lambda$ specifies the evolution of the magnetic field. We adopt a SBPL as spectral shape with $\alpha_s = -1/3$ and $\beta_s = 1.5$, an initial observed peak frequency of 100 keV and a thickness of the expanding shell that is constant in time. The dot-dashed line is the evolution expected in case of HLE without adiabatic cooling, assuming the same spectral shape and initial observed peak frequency. The error bars represent $1\sigma$ uncertainties, calculated via spectral fitting in XSPEC. In **b** we show the temporal evolution of normalized flux expected in case of adiabatic cooling. $\delta t_{obs} + 100$ s is the time measured from the peak of the decay shifted at 100 s, the typical starting time of the tail emission detected by XRT. We adopt the same parameters as in **a**, assuming $R_0 = 2 \times 10^{15}$ cm and $\Gamma = 100$. The dot-dashed line is the corresponding HLE model without accounting for adiabatic cooling. $\tau_{ad} = R_0/2c\Gamma^2$ indicates the timescale of adiabatic cooling, which is the same of HLE. The vertical error bars represent $1\sigma$ uncertainties and they are calculated via spectral fitting in XSPEC, while horizontal error bars represent the width of the time bin. In the legend we report the name of each GRB.

indicating that the spectral evolution becomes dominated by the emission at larger angles, rather than by adiabatic cooling in the jet core. Adiabatic cooling can also well reproduce the light curve of X-ray tails (Fig. 4b). For comparison, in the same plot, we show the light curve given by pure HLE, adopting the same value of $R_0$ and $\Gamma$.

In order to fully explore the parameter space of the adiabatic cooling model, we used a Monte Carlo Markov Chain (MCMC) algorithm for the parameter estimation. We consider the joint temporal evolution of flux and photon index and we find agreement of the model with data (see "Methods:" "Parameter

**Table 1 Results of the parameter estimation via MCMC, adopting the adiabatic cooling model.**

| GRB | $E_{peak}$ (keV) | $\lambda$ | $\tau_{ad}$ (s) |
|---|---|---|---|
| 090621A | $18^{+3}_{-2}$ | $2.11^{+0.56}_{-0.54}$ | $24.4^{+4.7}_{-3.0}$ |
| 100619A | >129 | $0.47^{+0.11}_{-0.07}$ | $0.3^{+1.0}_{-0.2}$ |
| 110102A | $46^{+15}_{-9}$ | $0.61^{+0.10}_{-0.10}$ | $5.8^{+1.9}_{-1.1}$ |
| 140512A | >323 | $0.48^{+0.04}_{-0.03}$ | $0.9^{+0.9}_{-0.4}$ |
| 161117A | $80^{+55}_{-21}$ | $0.69^{+0.10}_{-0.10}$ | $6.2^{+2.0}_{-2.3}$ |
| 170906A | $135^{+204}_{-53}$ | $0.66^{+0.10}_{-0.09}$ | $3.0^{+1.6}_{-1.5}$ |
| 180325A | >122 | $0.39^{+0.06}_{-0.05}$ | $0.8^{+1.3}_{-0.5}$ |
| 190604B | $54^{+227}_{-20}$ | $0.45^{+0.25}_{-0.15}$ | $3.5^{+2.6}_{-2.8}$ |

The confidence intervals and the lower limits represent the 16th, 50th, and 84th percentiles of the samples in the marginalized distributions (i.e., $1\sigma$ level of confidence).

estimation via MCMC" and Table 1). In Supplementary Figs. 9 and 10, we show for each burst the observed temporal evolution of photon index and normalized flux in comparison with the curves produced with 500 random draws from the posterior sample set of the MCMC. We obtain a value of $\lambda$ in the range 0.4–0.7 (except for 090621A which prefers $\lambda \sim 2$). On average, these values of $\lambda$ are smaller than those expected in an emitting region with a transverse magnetic field ($\lambda = 1$ or $\lambda = 2$ for a thick or a thin shell, respectively) or magnetic field in pressure equilibrium with the emitting particles ($\lambda = 4/3$ or $\lambda = 2$ for a thick or a thin shell, respectively[31]).

The typical timescale of adiabatic cooling $\tau_{ad} = R_0/2c\Gamma^2$, i.e., the observed time interval during which the radius doubles, is equal to the HLE timescale[13,32] and radically affects the slope of X-ray tails. Therefore, the comparison between the model and the observed light curves allows us to constrain the size $R_0$ of the emitting region as in HLE[33,34]. We find values in the range $0.3\,\mathrm{s} \lesssim \tau_{ad} \lesssim 24\,\mathrm{s}$. These values are quite larger than the typical duration of GRB pulses (<1 s[35]), which can be due to the following reason. For the spectral analysis to be feasible, we had to choose only tails that are long enough (to be divided down into a sufficient number of temporal bins). Moreover, the prompt emission is usually interpreted as a superposition of several emission episodes: the SD observed in XRT is likely dominated by the tails with the slowest decay timescales. Since, in our model, the decay timescale is $\tau_{ad} = R/2c\Gamma^2$, this could indicate that the emission radius of the pulses that dominate the tail is systematically larger than that of pulses that dominate the prompt emission. If this is the case, a lower magnetic field is also expected, which goes well along with the long radiative timescale and slow (or marginally fast) cooling regime, in agreement with our results. For the range of $\tau_{ad}$ obtained from the analysis, the corresponding range for the emission radius is $1.8 \times 10^{14}$ $(\Gamma/100)^2$ cm $\lesssim R_0 \lesssim 1.4 \times 10^{16}$ $(\Gamma/100)^2$ cm. A different prescription for adiabatic cooling has been suggested in the literature[30], in which the particle's momentum gets dynamically oriented transverse to the direction of the local magnetic field. In this case, HLE is the dominant contributor to the X-ray tail emission, which is again incompatible with the observed $\alpha - F$ relation.

**Extending the sample.** In order to further test the solidity of the $\alpha - F$ relation, we extend our analysis to a second sample of GRBs (composed by eight elements), which present directly a SD at the beginning of the XRT light curve, instead of an X-ray pulse (see Fig. 5a), often observed in early X-ray afterglows[8,9]. We require that the XRT SD is preceded by a pulse in the BAT light curve (the brightest since its trigger time). We add the data of this second sample to the $\alpha - F$ plot, estimating the peak flux by the extrapolation of the XRT light curve backwards to the peak time

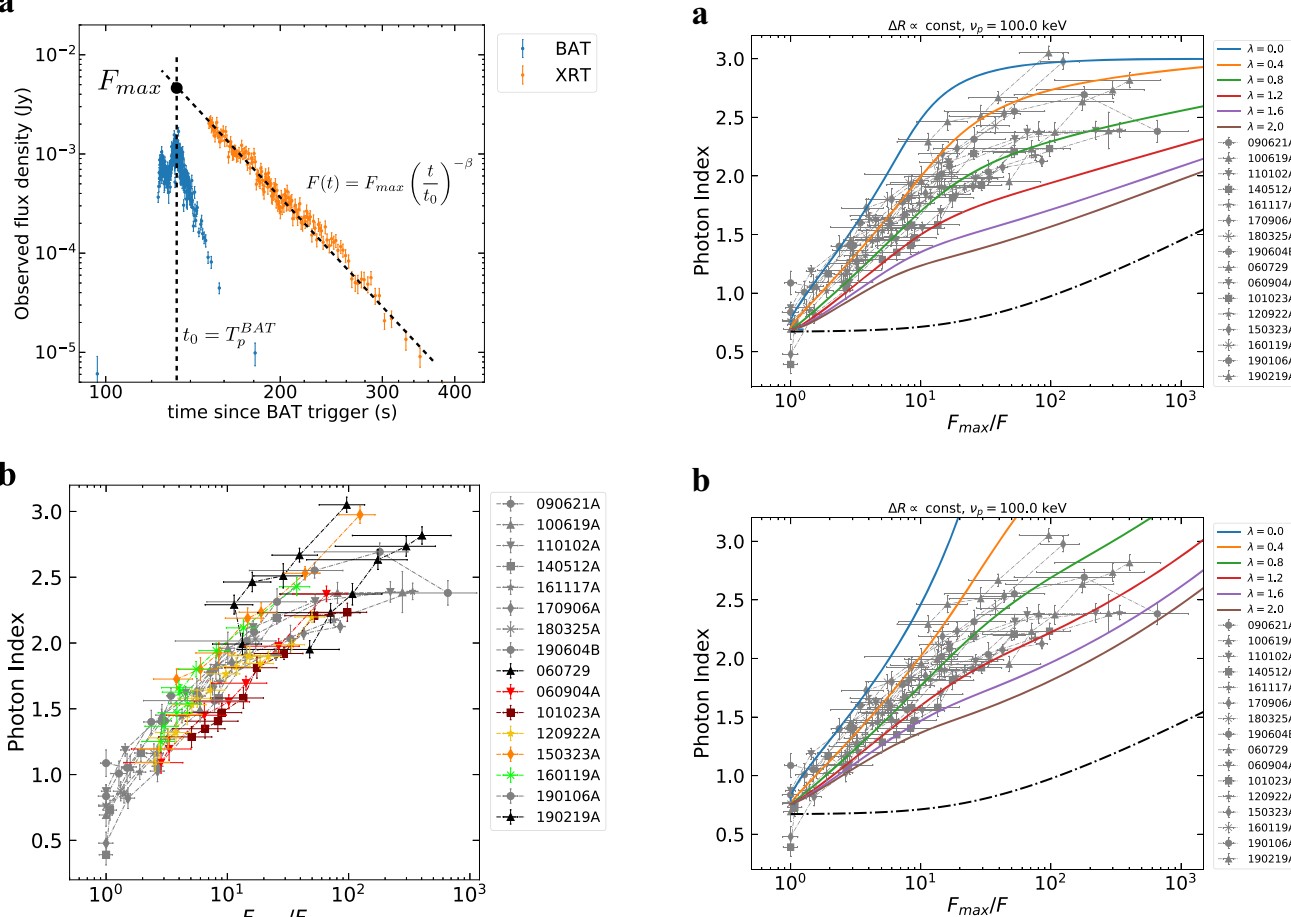

**Fig. 5 The steep decay phase and the correspondent spectral evolution for the extended sample.** In **a** we show an example of a light curve of an X-ray tail selected for our extended sample, taken from GRB 150323A. We report on the same plot the XRT (orange) and the BAT (blue) flux density at 1 and 50 keV, respectively. The peak flux $F_{max}$ is estimated extrapolating the X-ray tail back to the BAT peak. The error bars represent $1\sigma$ uncertainties and they are derived from the Swift archive. In **b** we show the spectral evolution of our extended sample of GRBs, which present a steep decay at the beginning of the XRT light curve, preceded by the brightest BAT pulse since the trigger time. The evolution of $\alpha$ lies on the same region of the plane occupied by the original sample, indicated in gray. The error bars represent $1\sigma$ uncertainties, calculated via spectral fitting in XSPEC. In the legend, we report the name of each GRB.

**Fig. 6 Spectral evolution expected in case of adiabatic cooling (solid lines) superimposed to the extended sample.** In **a**, the theoretical curves are computed considering adiabatic cooling and inefficient synchrotron cooling, taking also into account the effect of HLE. The value of $\lambda$ specifies the evolution of the magnetic field. We adopt a SBPL as spectral shape with $\alpha_s = -1/3$ and $\beta_s = 2.0$, an initial observed peak frequency of 100 keV and a thickness of the expanding shell that is constant in time. In **b** we show the spectral evolution expected in case of combined adiabatic cooling and mild synchrotron cooling. The adopted spectral shape is a SBPL plus an exponential cutoff. The initial peak frequency is 100 keV. The theoretical curves are computed taking also into account the effect of HLE. In both panels, the error bars represent $1\sigma$ uncertainties, calculated via spectral fitting in XSPEC, and the dot-dashed line is the evolution expected considering only HLE, assuming the same spectral shape and initial observed peak frequency. In the legend, we report the name of each GRB.

of the BAT pulse, under the assumption that BAT and XRT peaks were simultaneous (see "Methods:" "Extrapolation of $F_{max}$"). We find that these GRBs follow the overall $\alpha - F$ relation (Fig. 5b), confirming that a common physical process is governing the spectral evolution of X-ray tails. Adiabatic cooling is still capable of reproducing the data of this second sample (Fig. 6a), provided that we assume a slightly softer high-energy intrinsic spectrum ($\alpha \sim 3$ instead of $\alpha \sim 2.5$). Alternatively, the introduction of an exponential cutoff in the spectral shape at $\nu = \nu_c \sim \nu_m$ can also reproduce the data (see Fig. 6b), where $\nu_c$ and $\nu_m$ are the synchrotron characteristic frequencies. The cutoff is formed by a combined action of adiabatic cooling and mild synchrotron cooling (see Supplementary Note 4). We specify that the limited size of our samples is related to the selection requirements, which are necessary for an appropriate time-resolved spectral analysis. Thus our results are proved for X-ray tails firmly connected to prompt emission pulses.

## Discussion
The $\alpha - F$ relation, found in our analysis, requires a mechanism that produces the X-ray tails of GRBs with a unique law of flux decay and spectral softening. Although other scenarios cannot be ruled out, we find that adiabatic cooling of the emitting particles, together with a slowly decaying magnetic field, is the most plausible scenario able to robustly reproduce this relation. Our results suggest an efficient coupling between a slowly decaying magnetic field and the emitting particles. Our findings are generally in agreement with moderately fast and slow cooling regimes of the synchrotron radiation, which is able to reproduce the overall GRB spectral features[36]. In the adiabatic cooling scenario, most of the internal energy is not radiated away before the system substantially expands. If electrons are responsible for the emission, an extremely small magnetic field would be required[37–39], which is unrealistic for this kind of outflows. Protons radiating

through synchrotron emission can solve this problem[40]. Due to their larger mass, they radiate less efficiently than electrons, explaining why adiabatic cooling dominates the spectral evolution.

In conclusion, our results indicate that adiabatic cooling can play a crucial role for the collective evolution of the radiating particles in GRB outflows and consequently for the determination of spectral and temporal properties of prompt emission episodes. The coupling between particles and magnetic field ensures the intrinsic nature and hence the universality of this process, whose effects are independent of the global properties of the system, such as the luminosity of the GRB or the geometry of the jet.

## Methods

**Sample selection.** We define the SD segment[6–9] as the portion of the light curve that is well approximated by a power law, $F \propto t^{-\alpha}$ with $\alpha > 3$. Such criterion allows us to exclude a decay coming from a forward shock[10–12]. In order to determine the presence of a SD, we analyze the light curve of the integrated flux in the XRT (0.3–10) keV band.

From the Swift catalog[41] as of the end of 2019, we selected all GRBs with an XRT peak flux $F_p^{XRT} > 10^{-8}\,\mathrm{erg\,cm^{-2}\,s^{-1}}$. We selected the brightest pulses in order to have a good enough spectral quality as to perform a time-resolved spectral analysis. The peak flux is computed taking the maximum of $F(t_i)$, where $F(t_i)$ are the points of the light curve at each time $t_i$. Among these GRBs, we selected our first sample according to the following criteria:

(1) The XRT light curve shows at least one SD segment that is clean, i.e., without secondary peaks or relevant fluctuations.

(2) If we call $F_1$ and $F_2$, the fluxes at the beginning and at the end of the SD, respectively, we require that $\frac{F_1}{F_2} > 10$. This requirement is necessary to have a sufficient number of temporal bins inside the SD segment and therefore a well sampled spectral evolution.

(3) The beginning of the SD phase corresponds to a peak in the XRT light curve, such that we have a reliable reference for the initial time. We stress that the identification of the SD starting time in XRT is limited by the observational window of the instrument. This means that, if the XRT light curve starts directly with a SD phase, with no evidence of a peak, the initial reference time is possibly located before and its value cannot be directly derived.

(4) The XRT peak before the SD has a counterpart in BAT, whose peak is the brightest since the trigger time. This requirement is necessary to ensure that XRT is looking at a prompt emission episode, whose typical peak energy is above 100 keV. In a quantitative way, we define two times, $t_p$ and $t_{90}^{stop}$, where the first indicates the beginning of the peak that generates the SD, while the second is the end time of $T_{90}$[42], with respect to the trigger time. We require $t_{90}^{stop} > t_p$ in order to have an overlap between the last prompt pulses (monitored by BAT) and the XRT peak that precedes the SD phase. Namely, such requirement ensures that a considerable fraction of the energy released by the burst goes into the pulse that generates the X-ray tail.

It is possible that more than one peak is present in the XRT light curve, each with a following SD. In this case we consider only the SD after the brightest peak. If two peaks have a similar flux, we consider the SD with the larger value of $\frac{F_1}{F_2}$.

We then define a second sample of GRBs that satisfy the first two points listed before, but have a SD at the beginning of the XRT light curve, namely no initial peak preceding the SD is present. In addition, we require that a BAT pulse precedes the XRT SD and is the brightest since the trigger time. The BAT pulse enables us to constrain the starting time of the SD.

The selection criteria limit the size of our sample, but they are unavoidable to perform a well-targeted analysis of X-ray tails and to achieve robust conclusions about their origin.

**Time-resolved spectral analysis.** For each GRB, we divided the XRT light curve in several time bins, according to the following criteria:

(1) Each bin contains only data in windowed timing (WT) mode or in photon counting (PC) mode, since mixed WT + PC data cannot be analyzed as a single spectrum.

(2) Each bin contains a total number of counts $N_{bin}$ in the E = (0.3–10) keV band larger than a certain threshold $N_0$, which is chosen case by case according to the brightness of the source (see below). The definition of the time bins is obtained by an iterative process, i.e., starting from the first point of the light curve we keep including subsequent points until

$$N_{bin} = \sum_{t_n = t_i}^{t_f} N(t_n) > N_0 \qquad (1)$$

where $N(t_n)$ are the counts associated to each point of the light curve, while

$t_i$ and $t_f$ define the starting and ending time of the bin. Then the process is repeated for the next bins, until $t_f$ is equal to the XRT ending time. Due to the large range of count rates covered during a typical XRT light curve, the choice of only one value for $N_0$ would create an assembly of short bins at the beginning and too long bins toward the end. Therefore, we use one value of $N_0$ for bins in WT mode ($N_0^{WT}$) and a smaller value of $N_0$ for bins in PC mode ($N_0^{PC}$). In our sample, the SD is usually observed in WT mode, therefore we adjust $N_0^{WT}$ in order to have at least 4–5 bins inside the SD. A typical value of $N_0^{WT}$ is around 1500–3000, while $N_0^{PC}$ is around 500–1000. Using these values, we verified that the relative errors of photon index and normalization resulting from spectral analysis are below ~30%.

(3) For each couple $(N_i, N_j)$ of points inside the bin, the following relation must hold:

$$\frac{|N_i - N_j|}{\sqrt{\sigma_i^2 + \sigma_j^2}} < 5 \qquad (2)$$

where $\sigma_i$ and $\sigma_j$ are the associated errors. Such requirement avoids large flux variations within the bin itself.

(4) The duration of the bin is larger than 5 s, in order to avoid pileup in the automatically produced XRT spectra.

It is possible that condition 3 is satisfied only for a duration of the bin $T_{bin} < T_0$, while condition 2 is satisfied for $T_{bin} > T_0^*$, but $T_0^* > T_0$, meaning that they cannot be satisfied at the same time. In this case, we give priority to condition 3, provided that $N_{bin}$ is not much smaller than $N_0$.

Due to the iterative process that defines the duration of the bins, it is possible that the last points in WT and PC mode are grouped in a single bin with a too small $N_{bin}$, giving a too noisy spectrum. Therefore, they are excluded from the spectral analysis.

**Spectral modeling.** The spectrum of each bin is obtained using the automatic online tool provided by Swift for spectral analysis (see "Data availability"). Each spectrum is analyzed using XSPEC[43], version 12.10.1, and the Python interface PyXspec. We discard all photons with energy $E < 0.5$ keV and $E > 10$ keV. The spectra are modeled with an absorbed power law, and for the absorption, we adopted the Tuebingen–Boulder model[44]. If the GRB redshift is known, we use two distinct absorbers, one Galactic[45] and one relative to the host galaxy (the XSPEC syntax is tbabs*ztbabs*po). The column density $N_H$ of the second absorber is estimated through the spectral analysis, as explained below. On the other hand, if the GRB redshift is unknown, we model the absorption as a single component located at redshift $z = 0$ (the XSPEC syntax is tbabs*po) and also in this case the value of $N_H$ is derived from spectral analysis.

For the estimation of the host $N_H$, we consider only the late part of the XRT light curve following the SD phase. At late time with respect to the trigger, we do not expect strong spectral evolution, as verified in several works in the literature[46,47]. Therefore, for each GRB, the spectrum of each bin after the SD is fitted adopting the same $N_H$, which is left free during the fit. Normalization and photon index are also left free, but they have different values for each spectrum. We call $N_H^{late}$ the value of $N_H$ obtained with this procedure. In principle, the burst can affect the ionization state of the surrounding medium, but we assume that such effects are negligible and $N_H$ does not change dramatically across the duration of the burst[48]. Hence, we analyzed separately all the spectra of the SD using a unique value of $N_H = N_H^{late}$, which is fixed during the fit. Normalization and photon index, instead, are left free.

An alternative method for the derivation of $N_H$ is the fitting of all the spectra simultaneously imposing a unique value of $N_H$ that is left free. On the other hand, since $N_H$ and photon index are correlated, an intrinsic spectral evolution can induce an incorrect estimation of $N_H$. For the same reason we do not fit the spectra adopting a free $N_H$, since we would obtain an evolution of photon index strongly affected by the degeneracy with $N_H$.

In this regard, we tested how our results about spectral evolution depend on the choice of $N_H$. On average, we found that the fits of the SD spectra remain good (stat/dof $\lesssim 1$) for a variation of $N_H$ of about 50%. As a consequence, the photon index derived by the fit would change at most of 30%. Therefore the error bars reported in all the plots $\alpha - F$ are possibly underestimated, but even considering a systematic error that corresponds to ~30% of the value itself would not undermine the solidity of the results.

**Extrapolation of $F_{max}$.** We explain here how we extrapolated the $F_{max}$ for the GRBs of the second sample, for which the XRT light curve starts directly with a SD. We consider the peak time $T_p^{BAT}$ of the BAT pulse that precedes the SD. In the assumption that the SD starts at $T_p^{BAT}$, we can derive $F_{max}$ using the following procedure. We consider the (0.5–10) keV flux $F(t_i)$ for each bin time $t_i$ in the SD, derived from spectral analysis. Then we fit these points with a power law

$$F(t_i) = F_{max}\left(\frac{t_i}{t_0}\right)^{-s} \qquad (3)$$

with $s > 0$ and imposing that $t_0 = T_p^{BAT}$. Finally we derive the best fit value of $F_{max}$

with the associated $1\sigma$ error. The error of $F_{max}$ has a contribution coming from the error associated to $s$ and another associated to $t_0$, as well as from the assumption of a power law as fitting function. The value of $T_p^{BAT}$ is obtained fitting the BAT pulse with a Gaussian profile. Since usually the BAT pulse can have multiple sub-peaks and taking also into account possible lags between XRT and BAT peaks, we adopt a conservative error associated to $T_p^{BAT}$ equal to 5 s.

**HLE from infinitesimal duration pulse**. We assume that an infinitesimal duration pulse of radiation is emitted on the surface of a spherical shell, at radius $R_0$ from the center of the burst. Such treatment implicitly assumes particles that cool on timescales much smaller than the dynamical timescales. Therefore, all the X-ray tail emission is dominated by photons departed simultaneously from the last emitting surface. The jet has an aperture angle $\vartheta_j$ and it expands with a bulk Lorentz factor $\Gamma$. We assume also that the comoving spectrum is the same on the whole jet surface. The temporal evolution of the observed flux density is given by ref. [49]

$$F_\nu(t_{obs}) \propto S_{\nu'}(\nu/\mathcal{D}(\vartheta))\mathcal{D}^2(\vartheta)\cos(\vartheta) \tag{4}$$

with $S_{\nu'}(\nu/\mathcal{D}(\vartheta))$ the comoving spectral shape, $\mathcal{D}(\vartheta)$ the Doppler factor, and $\vartheta$ the angle measured from the line of sight, which is assumed to coincide with the jet symmetry axis. The observer time $t_{obs}$ is related to the angle $\vartheta$ through the following formula:

$$t_{obs}(\vartheta) = t_{em}(1 - \beta\cos\vartheta) \tag{5}$$

where $t_{em}$ is the emission time. Equation (4) is valid for $\vartheta < \vartheta_j$, while for $\vartheta > \vartheta_j$ the emission drops to zero. This implies that for $t_{obs} > t_{em}(1 - \beta\cos\vartheta_j)$, the flux drops to zero. At each time $t_{obs}(\vartheta)$, the observer receives a spectrum that is Doppler shifted by a factor $\mathcal{D}(\vartheta)$ with respect to the comoving spectrum. If the comoving spectrum is curved, i.e., if $\frac{d^2}{d\nu'^2}S_{\nu'} \neq 0$, then also the photon index is a function of time[16]. The shape of the resulting curve $\alpha - F$ is determined only by the spectral shape and the comoving peak frequency $\nu'_p$, while it is independent on the emission radius $R_0$ and the bulk Lorentz factor $\Gamma$.

We notice that the observed photon index goes from 0.5–1.0 up to 2.0–2.5, consistent with the slopes of a synchrotron spectrum before and after the peak frequency. Indeed for a population of particles with an injected energy distribution $N(\gamma) \propto \gamma^{-p}$ that has not completely cooled, the expected shape of the spectrum is $F_\nu \sim \nu^{1/3}(\alpha = 2/3)$ for $\nu < \nu_c$ and $F_\nu \sim \nu^{-p/2}(\alpha = p/2 + 1)$ for $\nu > \nu_m \gtrsim \nu_c$. Hereafter, if not otherwise specified, we assume a spectral shape given by a smoothly broken lower law, which well approximates the synchrotron spectrum below and above the peak frequency. The form of the adopted spectral shape is

$$S_\nu \propto \frac{1}{\left(\frac{\nu}{\nu_0}\right)^{\alpha_s} + \left(\frac{\nu}{\nu_0}\right)^{\beta_s}} \tag{6}$$

with $\alpha_s = -1/3$ and $\beta_s = 1.5$. The peak frequency $\nu_p$ of the energy spectrum $\nu S_\nu$ is related to $\nu_0$ through the following relation:

$$\nu_p = \left(-\frac{2+\alpha_s}{2+\beta_s}\right)^{\frac{1}{\alpha_s - \beta_s}}\nu_0 \tag{7}$$

At each arrival time, we compute the flux and the photon index in the XRT band using Eq. (4). In particular, the XRT flux is given by

$$F_{0.5-10keV}(t_{obs}) = \int_{0.5\ keV/h}^{10\ keV/h} F_\nu(t_{obs})d\nu \tag{8}$$

where $h$ is Planck's constant, while the photon index is computed as[16,24]

$$\alpha(t_{obs}) = 1 - \frac{\log[F_{\nu=10keV/h}(t_{obs})/F_{\nu=0.5keV/h}(t_{obs})]}{\log(10\ keV/0.5\ keV)} \tag{9}$$

This method for the evaluation of photon index is valid in the limit of a spectrum that can be always approximated with a power law as it passes through the XRT band, which is the case for typical prompt emission spectra.

In addition to the SBPL, we test HLE also using other spectral shapes. We first adopt a Band function[23] with the following form:

$$B(\epsilon) = \begin{cases} \epsilon^{1+\alpha_s}e^{-\epsilon} & \epsilon < \alpha_s - \beta_s \\ (\alpha_s - \beta_s)^{\alpha_s-\beta_s}e^{-\alpha_s+\beta_s}\epsilon^{1+\beta_s} & \epsilon > \alpha_s - \beta_s \end{cases} \tag{10}$$

where $\epsilon = \nu/\nu_0$. In this case, the peak of the energy spectrum is at $\nu_p = (2+\alpha_s)\nu_0$. The resulting spectral evolution is very similar to the case of SBPL, as visible in Supplementary Fig. 12a.

As a final attempt, we use synchrotron spectrum emitted by a population of particles with an initial energy distribution $N(\gamma) \propto \gamma^{-P}$. Synchrotron is considered the dominant radiative process in prompt emission of GRBs[4,36]. In the fast cooling regime, the particle distribution becomes

$$N(\gamma) \propto \begin{cases} \gamma^{-2} & \gamma_c < \gamma < \gamma_m \\ \gamma^{-(p+1)} & \gamma > \gamma_m \end{cases} \tag{11}$$

The only three parameters that define the shape of the synchrotron spectrum are

$\nu_m \propto \gamma_m^2$, $\nu_c \propto \gamma_c^2$, and $p$. For the computation of the spectrum we use[50]

$$F_\nu \propto \int_{\gamma_c}^\infty P(\nu,\gamma)N(\gamma)d\gamma \tag{12}$$

with

$$P(\nu,\gamma) \propto B\left[\left(\frac{\nu}{\nu_{ch}}\right)\int_{\frac{\nu}{\nu_{ch}}}^\infty K_{5/3}(x)dx\right], \quad \nu_{ch} \propto \gamma^2 B \tag{13}$$

where $B$ is the magnetic field and $K_{5/3}(x)$ is the modified Bessel function of order 5/3. The resulting spectral evolution for values of $\nu_m/\nu_c = 1$ and $\nu_m/\nu_c = 10$ is reported in Supplementary Fig. 11. A value of $\nu_m/\nu_c \sim 1$ is expected in the marginally fast cooling regime[37–39], which is favored by broad-band observations of GRB prompt spectra[51–56]. Finally, we test how the sharpness of the spectral peak can affect our results. In particular we consider again a SBPL and we generalize the formula adding a sharpness parameter $n$

$$S_\nu^{(n)}\left(\frac{\nu}{\nu_0}\right) \propto \left[\frac{1}{\left(\frac{\nu}{\nu_0}\right)^{n\alpha_s} + \left(\frac{\nu}{\nu_0}\right)^{n\beta_s}}\right]^{1/n} \tag{14}$$

where larger values of $n$ correspond to sharper spectral peaks. As visible in Supplementary Fig. 12b, where we have adopted $n = 4$, the shape of the curves becomes flatter at the beginning and at the end of the decay, but with no substantial steepening of the intermediate part. This is attributable to HLE that imposes an evolution of the observed peak frequency like $t_{obs}^{-1}$. Thus, while the initial and final values of photon index are dictated by the spectral shape, the steepness of the transition from the initial to the final value is governed by HLE and is independent on the spectral shape. In conclusion, no one of the alternative spectral shapes that we tested is able to reconcile HLE with the observed spectral evolution.

We finally test how the $\alpha - F$ relation changes if we assume a structured jet with an angle-dependent comoving spectrum. In particular, we consider a spectral peak energy that is nearly constant inside an angle $\vartheta_c$ (measured with respect to the line of sight) and starts to decrease outside it. Regardless of the choice of the specific law for the angular dependence (e.g., Gaussian or power law), the HLE can reproduce the $\alpha - F$ relation only if all the analyzed GRBs have a fine-tuned value of $\vartheta_c < 1°$. Such small value of $\vartheta_c$, on the other hand, would imply a very short SD, in contradiction with observations.

**Adiabatic cooling**. In this section, we derive the effect of adiabatic cooling of the emitting particles[57] on the light curve and the spectral evolution of X-ray tails. We assume that the emission is dominated by a single species of particles that can be treated as a relativistic gas in adiabatic expansion. We assume also that there is no interaction with other species of particles. If the particles are embedded in a region of comoving volume $V'$, an adiabatic expansion satisfies the equation

$$\langle\gamma\rangle^3 V' = const \tag{15}$$

where $\langle\gamma\rangle$ is the average Lorentz factor of the emitting particles in the comoving frame. The last equation is valid in the limit in which the adiabatic cooling timescale is smaller than the cooling time of other radiative processes, such as synchrotron or inverse Compton. Namely, particles radiate only a negligible fraction of their internal energy during the expansion of the system. Regarding the radial dependence of the volume $V'$, we distinguish two cases:

(1)  thick shell, with a comoving width $\Delta R'$ that does not evolve with time, hence $V' \propto R^2\Delta R' \propto R^2$

(2)  thin shell, with a comoving width $\Delta R'$ that evolves linearly with $R$, hence $V' \propto R^2\Delta R' \propto R^3$.

We assume that the dominant radiative process is synchrotron. The evolution of the spectrum in the observer frame is therefore fully determined once we know how the spectrum normalization $F_{\nu_p}$ and the peak frequency $\nu_p$ evolve in time. These two quantities, under the assumption of constant total number of emitting particles and constant bulk Lorentz factor $\Gamma$, take the following form:

$$F_{\nu_p} \propto B, \quad \nu_p \propto \langle\gamma\rangle^2 B \tag{16}$$

where $B$ is the magnetic field (assumed tangled) as measured by a comoving observer. As described in the main text, we adopt the following parametrization for the magnetic field

$$B = B_0\left(\frac{R}{R_0}\right)^{-\lambda} \tag{17}$$

where $\lambda \geq 0$, under the reasonable assumption that magnetic field has to decrease or at most remain constant during the expansion. The value of $R_0$ corresponds to the radius where particles are injected, namely when adiabatic cooling starts to dominate. We use the integration along the EATS to compute the evolution of flux, as done for HLE from finite-duration pulse (see Supplementary Note 1), with the only difference that in this case the emission never switches off. The final form of

**Table 2 Comparison of best fit statistics between adiabatic cooling (AC) and HLE, adopting a SBPL, a Band function or a synchrotron spectrum, using the Akaike information criterion (AIC).**

| GRB | $(\Delta_{AIC})\mid_{SBPL}$ | $(\Delta_{AIC})\mid_{Band}$ | $(\Delta_{AIC})\mid_{Sync}$ |
|---|---|---|---|
| 090621A | 8 | 6 | 9 |
| 100619A | 69 | 66 | 67 |
| 110102A | 145 | 141 | 145 |
| 140512A | 193 | 190 | 43 |
| 161117A | 132 | 124 | 133 |
| 170906A | 148 | 135 | 145 |
| 180325A | 80 | 76 | 91 |
| 190604B | 61 | 59 | 65 |

The large values of $(\Delta_{AIC})\mid_{spectrum} = (AIC_{HLE} - AIC_{AC})\mid_{spectrum}$ indicate that, regardless of the assumed spectral shape, the HLE from efficiently cooled particles is strongly disfavored with respect to the adiabatic cooling model.

the integral is

$$F_\nu(t_{obs}) \propto \int_0^{\vartheta_j} S_{\nu'}(\nu/\mathcal{D}(\vartheta)) \left(\frac{R(\vartheta, t_{obs})}{R_0}\right)^{-\lambda} \mathcal{D}^3(\vartheta) \sin\vartheta \cos\vartheta d\vartheta \quad (18)$$

where the factor $(R/R_0)^{-\lambda}$ comes from $I'_{\nu'_p} \propto B$, while $\nu'$ evolves in time according to Eq. (16).

**Parameter estimation via MCMC.** In order to fully explore the parameter space of the adiabatic cooling model, we performed a MCMC, using the emcee algorithm[58]. The setup of our analysis is described in the following points:

(1) The model contains as free parameters $E_p$, $\lambda$ and $\tau_{ad} = R/2c\Gamma^2$, which are the peak energy at the beginning of the SD, the decay index of the magnetic field, and the adiabatic timescale. The inclusion of $\Gamma$ as free parameter returns a flat posterior distribution, indicating that the model is insensitive to it. Therefore we performed the analysis fixing $\Gamma = 100$.

(2) The MCMC in performed jointly for flux and photon index evolution. The adopted likelihood is

$$\log(\mathcal{L}) = -\frac{1}{2}\sum_n \left[\frac{(\phi_n - \bar\phi(t_n))^2}{s_{\phi,n}^2} + \ln\left(2\pi s_{\phi,n}^2\right)\right] - \frac{1}{2}\sum_n \left[\frac{(\alpha_n - \bar\alpha(t_n))^2}{s_{\alpha,n}^2} + \ln\left(2\pi s_{\alpha,n}^2\right)\right] \quad (19)$$

where $\phi = F/F_{max}$, $\alpha$ is the photon index, and with $\bar\phi$, $\bar\alpha$ we indicate the value predicted by the model at each time $t_n$. Moreover,

$$s_{\phi,n}^2 = \sigma_{\phi,n}^2 + f_\phi \cdot \bar\phi(t_n), \quad s_{\alpha,n}^2 = \sigma_{\alpha,n}^2 + f_\alpha \cdot \bar\alpha(t_n) \quad (20)$$

where $\sigma_\phi$ and $\sigma_\alpha$ are the errors, while $f_\phi$ and $f_\alpha$ are introduced to take into account possible underestimation of the errors. Since keeping $f_\alpha \neq 0$ leads to a posterior distribution of $f_\alpha$ peaked around $\sim 10^{-7}$, the parameter estimation was performed fixing $f_\alpha = 0$. Instead, we keep $f_\phi \neq 0$ taking into account that the error on the flux resulting from the fitting of time-averaged spectrum may not represent the true flux error over the time bin.

(3) The MCMC runs until the number of steps exceeds 100 times the autocorrelation time (its maximum) and the averaged autocorrelation time (over 100 steps) becomes constant within 1% accuracy. The burn-in is chosen as twice of autocorrelation time. As an example, we show in Supplementary Fig. 7, the evolution of the autocorrelation time as a function of the steps.

The resulting parameter estimation is summarized in Table 1. The uncertainties are reported based on the 16th, 50th, and 84th percentiles of the samples in the marginalized distributions ($1\sigma$ level of confidence). An example of corner plot obtained via MCMC is shown in Supplementary Fig. 8. In Supplementary Figs. 9 and 10, we show for each burst the observed temporal evolution of photon index and normalized flux in comparison with the curves produced with 500 random draws from the posterior sample set of the MCMC.

We performed an analogous MCMC analysis adopting the model of HLE from an instantaneous emission. However, the algorithm is unable to converge, demonstrating that the model cannot successfully match with the observations. The only way to obtain converged chains by this model is to admit extreme and unrealistic values of $f_\phi$, of the order $10^4$–$10^8$. The only exception is GRB 090621, which can be fitted by HLE alone. This is the only case where it is meaningful to compute the Bayes factor between HLE and AC, which results to be $\sim 200$. Thus we prove that the adiabatic cooling model is strongly preferred for all the analyzed cases.

The model comparison (adiabatic cooling + HLE against HLE only) is also done assuming different spectral shapes: SBPL, Band, and synchrotron. The

spectral parameters are the same as those adopted before. For synchrotron, we use $\nu_c = \nu_m$ (the case $\nu_c \neq \nu_m$ does not improve the goodness of fit). In order to compare the goodness of fit of the two models, we used the Akaike information criterion (AIC), which is defined as $AIC = 2k - 2\ln(\mathcal{L})$, where $k$ in the number of parameters of the model, and $\mathcal{L}$ is the best fit likelihood, that is, $2\ln(\mathcal{L}) = -\chi^2$. In Table 2, we show the value of $\Delta_{AIC} = AIC_{HLE} - AIC_{AC}$ for each spectral shape. For all cases, the adiabatic cooling is significantly favored with respect to HLE.

## Data availability
Raw data are public and available in the UK Swift Science Data Centre at the University of Leicester. The light curve data are taken at this link: https://www.swift.ac.uk/xrt_curves/GRB_ID/flux.qdp where GRB_ID is the GRB observation ID. The spectra are obtained at this link: https://www.swift.ac.uk/xrt_spectra/addspec.php?targ=GRB_ID where GRB_ID is the ID number of the GRB. The details of the automatic spectral analysis can be found here: https://www.swift.ac.uk/xrt_spectra/docs.php. Derived data are available from the corresponding author on request.

## Code availability
Codes used to produce the plots in this paper are available in this public repository: https://github.com/samueleronchini/Nature_communications. XSPEC and PyXspec are freely available online at the following links: https://heasarc.gsfc.nasa.gov/xanadu/xspec/ https://heasarc.gsfc.nasa.gov/docs/xanadu/xspec/python/html/index.html.

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

## Acknowledgements

The research leading to these results has received funding from the European Union's Horizon 2020 Programme under the AHEAD2020 project (Grant agreement n. 871158). G. Ghir. acknowledges the support from the ASI-Nustar Grant (1.05.04.95). M.B., P.D., and G.G. acknowledge support from PRIN-MIUR 2017 (Grant 20179ZF5KS). G.O. acknowledges financial contribution from the agreement ASI-INAF n.2017-14-H.0. S.A. acknowledges the PRIN-INAF "Towards the SKA and CTA era: discovery, localization, and physics of transient sources" and the ERC Consolidator Grant "MAGNESIA" (nr. 817661). M.G.B. and P.D. acknowledge ASI Grant I/004/11/3. O.S.S. acknowledges the INAF-Prin 2017 (1.05.01.88.06) and the Italian Ministry for University and Research Grant "FIGARO" (1.05.06.13) for support. G.O. and S.R. are thankful to INAF—Osservatorio Astronomico di Brera for kind hospitality during the completion of this work. This work made use of data supplied by the UK Swift Science Data Centre at the University of Leicester.

## Author contributions

S.R., G.O. and M.B. conceived the idea behind the paper. S.R. performed the sample selection and the spectral analysis, with the help of M.G.B. and P.D. S.R. and G.O. implemented the models and related numerical calculations. S.R. and G.O. developed the codes to compare models and data. S.R., G.O., M.B., S.A., M.G.B., F.B., S.D., P.D., G.Ghir., G.Ghis., M.E.R. and O.S.S. contributed in the discussion and in the interpretation of the results, as well as in writing and revising the manuscript.

## Competing interests

The authors declare no competing interests.
