## [Peer Review File · Nature Communications]

REVIEWER COMMENTS

Reviewer #1 (Remarks to the Author):

I read with interest the manuscript titled "Unveiling the origin of steep decay in γ -ray bursts" by Samuele Ronchini and Collaborators. In this paper, the authors claimed to discover an α -F relation during the steep decay phase of the X-ray light curves of some bright GRBs. The authors further argued that such relation is not compatible with the previous-thought high-latitude effect, and the proposed adiabatic cooling process can play a role in shaping the X-ray tails' temporal and spectral properties. I found the paper's general idea is novel and the authors discussed in detail the cooling process on top of the existing effects, which provides a useful reference to the field. However, I am not fully convinced by the approach the author used to compare theory with observations. The following comments should be taken into account by the authors before I can recommend it for publication in Nature Communications:

(1) The α -F correlation, although seemingly attractive, is not at all surprising. This is expected because flux drops and spectra soften during the steep decay phase when time increases. That said, I agree it is interesting to check if such correlations among a big sample share some common features, e.g., (same slopes), or not, which, unfortunately, was not discussed in the paper.

(2) The paper's key point is shown in Figure 3, in which the authors argue that adiabatic cooling can well reproduce the light curve of X-ray tails and α -F relation with some values of λ . However, I didn't find any "goodness of fit" here. The authors tried to explain the observed trends of a large sample as a whole using the adiabatic cooling (+ HLE) effect. For example, the curve with $\lambda = 0.4$ seems to be consistent with the observation's overall trends, but it is not guaranteed for individual bursts and not proved that the adiabatic cooling (+HLE) model can well produce the data.

(3) I thus suggest the author do the fits on a single burst basis. For each burst of their sample, the authors may refer to Figure 2 of Zhang, 2012, ApJ, 756, 190, and perform the model (e.g., put λ as a free parameter in their model) fit on both the flux and spectral evolution. Such a fit will provide direct evidence of the argument of the paper.

(4) For other models that the authors disfavored, such as HLE emission from an infinitesimal duration pulse with an SBPL/synchrotron/Band function spectrum, I suggest the authors do an apple-to-apple comparative fit to prove to what extent adiabatic cooling is needed.

(5) Figure 4, any explanation between slopes of BAT and XRT tails?

(6) Line 222, I believe alpha should be > 3 instead of > 2 .

(7) Line 261-262: "threshold N_0 is chosen case by case according to the brightness of the source." This is vague. At least some specific numbers should be provided.

(8) Line 318: I suggest replacing beta with a different symbol, as it is already used in Eq 2.

(9)_Line 431-432: I believed that "overall velocity and spectral distribution of mini-jets be the same for all the GRBs...to reproduce the $\alpha - F$ relation" is overstated. This has to be calculated, fitted and proved in more detail.

Reviewer #2 (Remarks to the Author):

The paper "Unveiling the origin of steep decay in gamma-ray bursts" by Ronchini et al. provides a novel way of understanding the origin of the X-ray steep decay in GRBs. The paper is well written, concise and provides a comprehensive study on understanding the X-ray steep decay in GRBs, which is an important problem, since it is deeply connected to the nature of the prompt GRB emission process. There are some points that could be improved in the paper, but overall the paper is quite interesting, well written and certainly advances our study of these enigmatic phenomena.

There are two points that require some attention. These two points have already been addressed by the authors but could be emphasized/clarified. The first one and more important is the fact that the

relevant timescale obtained in their analysis is on the order of a second to a few seconds. This is quite large compared to the typical duration of GRB pulses, which last much less than a second. This should be emphasized more clearly in the paper as an inconsistency of their model. The second point is the mention toward the end of the Introduction of the alpha – Flux relation. Although the results seem to show that this relation is robust for their samples, it remains to be seen and studied if it applies to all bursts with available X-ray steep decay data.

There are some other points that could be addressed, although some of these are suggestions: 1. Since the authors are using the peak in BAT, can the analysis of the authors extend to the BAT data as well and understand the decay seen in the BAT pulse considered in the paper? In this case, can the model explain the sharp peak in the BAT pulse and the broader peak in the X-ray data (see, e.g., Fig. 1 left panel)? 2. The model yields a prediction for the peak energy as a function of time as it crosses the X-ray band. Does the time-resolved XRT analysis show this? This is probably hard to see due to the narrowness of the XRT band, but it is worth studying. 3. The inset of the right panel of Figure 2 refers to the 3 spectra, not only the blue spectra: The current figure makes the reader think that the inset only zooms-in into the blue line. This can be clarified. 4. The authors mention that the shell thickness choice is irrelevant since it does not change the result significantly, perhaps a figure addressing this could be included in the supplementary material. 5. There is a minor typo in equation (18), (dR/dt) should be squared, however, equation (19) is correct.

Reviewer #3 (Remarks to the Author):

In this paper, the authors analyzed X-ray data in the steep decay phase, which has been interpreted as emission from a higher latitude. This paper shows a relation between the X-ray flux and photon-spectral index. The photon index evolves fast with decaying flux. This spectral evolution is significantly faster than expected in the high latitude emission model. This unexpected finding has great merit to be published. This might change the standard picture of GRB emission mechanism. However, the proposed model to explain their results in this paper has several unnatural points. I feel that the claim "evidence of adiabatic cooling" is too strong.

The revealed thing in this paper is the evolution of the photon index with the decaying flux. This behavior is consistent with the model shifting the spectral shape to lower energy and lower flux with the same functional shape. The unchanged spectral shape is consistent with the adiabatic cooling. However, the shift of the spectral peak energy or unchanged-but-shifted spectral shape itself is not verified. The adiabatic cooling model is not unique solution for the index evolution in this paper. Other types of non-trivial spectral-flux evolution can agree with the results in this paper.

As shortly mentioned in 3. Conclusions, if the adiabatic cooling is the dominant cooling process, the radiative cooling timescale should be longer than the dynamical timescale. The electrons with average Lorentz factor $\langle \gamma \rangle$ are roughly responsible for the spectral peak energy $\sim \text{MeV}$. Achieving this peak energy and long radiative cooling timescale are compatible? Such discussion is lacking.

Or the authors strongly support proton synchrotron model? If so, the authors should emphasize proton synchrotron rather than adiabatic cooling. In this case, is the jet energy dominated by protons or magnetic field? How much is the energy conversion efficiency to photons?

The HLE model and adiabatic cooling model are compared. However, while the cooling process for the adiabatic cooling model is calculated, a sudden shutdown of the emission without spectral softening is assumed in the HLE model. This may be not even treatment. Even in the HLE model, the spectral shape evolves in the decaying phase with particle cooling and decaying particle injection, probably showing softening. The late time such emission from high latitude may contribute to the emission.

I can not understand the phrase in abstract "evidence of ... efficient energy exchange between the emitting particles in the relativistic outflows of GRBs." Adiabatic means no energy exchange. Particle energy is not efficiently released as photons. So most of the energy may remain in the jet not being released.

Re-acceleration/slow heating of electrons is rejected in this paper. This statement is also based on the sudden shutdown assumption. Even in such models, the spectra may show gradual softening with decaying re-acceleration or heating rate (e.g. Asano & Terasawa 2105). This may show a similar behavior to the adiabatic cooling model.

Point by point reply to reviewers

We thank the reviewers for their constructive comments. They raised points which were very relevant to improve our manuscript. We report their comments and our reply is denoted with “R”.

1 Reviewer 1 (Remarks to the Author)

I read with interest the manuscript titled ”Unveiling the origin of steep decay in gamma-ray bursts” by Samuele Ronchini and Collaborators. In this paper, the authors claimed to discover an alpha-F relation during the steep decay phase of the X-ray light curves of some bright GRBs. The authors further argued that such relation is not compatible with the previous-thought high-latitude effect, and the proposed adiabatic cooling process can play a role in shaping the X-ray tails’ temporal and spectral properties. I found the paper’s general idea is novel and the authors discussed in detail the cooling process on top of the existing effects, which provides a useful reference to the field. However, I am not fully convinced by the approach the author used to compare theory with observations. The following comments should be taken into account by the authors before I can recommend it for publication in Nature Communications:

1.1) The alpha-F correlation, although seemingly attractive, is not at all surprising. This is expected because flux drops and spectra soften during the steep decay phase when time increases. That said, I agree it is interesting to check if such correlations among a big sample share some common features, e.g., (same slopes), or not, which, unfortunately, was not discussed in the paper.

R: The novelty of our work resides in the universality of the correlation, i.e., all the tails in our sample obey an almost identical relation between the flux and the photon index. We agree that a simultaneous flux drop and spectral softening is expected and already found in the literature. However the existence of a common relation for all GRBs, independent on individual properties like, e.g., peak time, tail duration or intrinsic luminosity, was neither predicted nor guaranteed *a priori*, given the diversity of GRB properties.

In this regard it is instructive to note that, in the case of a spectrum that is rigidly shifted to lower energies and whose normalization decreases in time as well, the photon index is a function of the normalized flux only. Indeed, under these assumptions, the flux density at a given frequency is determined by the peak frequency $\nu_p(t)$ and the corresponding peak flux $F_p(t)$, i.e. $F_\nu = F_\nu(\nu_p)$. Moreover the photon index α in a given frequency band represents the local derivative of the photon spectrum and hence also $\alpha(t) = \alpha(\nu_p)$. Therefore, since both F_ν and α are only functions of ν_p , then the $\alpha - F$ relation is uniquely determined by the process that governs the spectral evolution and does not depend on additional parameters (e.g. the bulk Lorentz factor or the radius of the outflow).

Regarding the possibility of checking the correlation on a larger sample, please refer to the discussion in point 2.2, where the second reviewer raises a similar issue. Two clarifying sentences about this point are added at the end of the section “Extending the sample” and in the Methods section “Sample selection”.

1.2) The paper’s key point is shown in Figure 3, in which the authors argue that adiabatic cooling can well reproduce the light curve of X-ray tails and alpha-F relation with some values of λ . However, I didn’t find any ”goodness of fit” here. The authors tried to explain the observed trends of a large sample as a whole using the adiabatic cooling (+ HLE) effect. For example, the curve with $\lambda = 0.4$ seems to be consistent with the observation’s overall trends, but it is not guaranteed for individual bursts and not proved that the adiabatic cooling (+HLE) model can well produce the data.

R: This point has been addressed with the following discussion.

1.3) I thus suggest the author do the fits on a single burst basis. For each burst of their sample, the authors may refer to Figure 2 of Zhang, 2012, ApJ, 756, 190, and perform the model (e.g., put λ as a free parameter in their model) fit on both the flux and spectral evolution. Such a fit will provide direct evidence of the argument of the paper.

R: We followed the suggestion of the reviewer, fitting our model (adiabatic cooling+HLE) to each burst of our sample, adopting the following procedure. First we fit the $\alpha - F$ relation and determine the best fit values of λ and the initial peak frequency ν_p , adopting a SBPL spectral shape with the same spectral slopes as used in the paper (the choice of different spectral shapes is discussed in the next point). Then we fit the light curve fixing λ and ν_p to the values found before, so that the only free parameter of the fit is the adiabatic time scale τ_{ad} . Best fits are obtained by minimizing the χ^2 . The light curves are taken from the Swift archive, with a 10 s binning. The results are reported in Tab. 1, confirming the agreement of our model with the data. In two cases, the reduced χ^2 is significantly larger than one either in the $\alpha - F$ relation fitting (GRB 140512A) or in the light curve fitting (GRB 110102A). In the first case, this is due to large residuals in the first points, due to the very hard photon index in correspondence of $F_{max}/F = 1$, which is incompatible with the adopted spectral shape, whose low energy photon index is $2/3$. In the second case, it is due to a deviation of the light curve shape from a simple power law.

We added the description of the fit results in the section “Adiabatic cooling”, along with the table 1.

1.4) For other models that the authors disfavored, such as HLE emission from an infinitesimal duration pulse with an SBPL/synchrotron/Band function spectrum, I suggest the authors do an apple-to-apple comparative fit to prove to what extent adiabatic cooling is needed.

R: In order to compare the adiabatic cooling (AC)+HLE model against the HLE-only model, we performed a one by one fit and compared the goodness of fit. For both models, three spectral shapes are tested: SBPL, Band and Synchrotron. The spectral parameters are the same as those adopted in the paper. For Synchrotron, we use $\nu_c = \nu_m$ (the case $\nu_c \neq \nu_m$ does not improve the goodness of fit). In order to compare the goodness of fit of the two models, we used the Akaike Information Criterion (AIC), which is defined as $AIC = 2k - 2\ln(\mathcal{L})$, where k is the number of parameters of the model, and \mathcal{L} is the best fit likelihood, that is, $2\ln(\mathcal{L}) = -\chi^2$. In Tab. 2 the best fit statistics and the value of $\Delta_{AIC} = AIC_{HLE} - AIC_{AC}$ are shown. For all cases, the AC is

name	λ	ν_p	χ^2/dof	dof	τ_{ad}	χ^2/dof	dof
	-	keV			s		
090621A	$1.79^{+0.06}_{-0.01}$	21^{+1}_{-2}	1.13	4	$22.8^{+0.54}_{-0.83}$	0.43	12
100619A	$0.41^{+0.12}_{-0.02}$	432^{+21}_{-61}	0.35	5	$0.78^{+0.10}_{-0.10}$	1.50	10
110102A	$0.67^{+0.14}_{-0.01}$	30^{+1}_{-3}	0.50	4	$13.5^{+0.20}_{-0.10}$	6.25	25
140512A	$0.67^{+0.03}_{-0.01}$	297^{+1}_{-20}	10.3	4	$4.8^{+0.4}_{-0.3}$	0.74	12
161117A	$0.70^{+0.08}_{-0.01}$	137^{+1}_{-11}	0.80	7	$8.0^{+0.2}_{-0.10}$	1.02	29
170906A	$0.71^{+0.04}_{-0.02}$	425^{+14}_{-34}	1.91	9	$4.0^{+0.2}_{-0.1}$	0.31	11
180325A	$0.59^{+0.07}_{-0.03}$	196^{+11}_{-33}	0.43	3	$5.6^{+0.3}_{-0.3}$	0.28	7
190604B	$0.64^{+0.04}_{-0.02}$	44^{+2}_{-3}	0.30	4	$16.7^{+1.0}_{-0.6}$	0.62	4

Table 1: Best fit parameters of single bursts, adopting adiabatic cooling+HLE.

significantly favored with respect to HLE.

We added these results in the Methods section “Adiabatic cooling”, along with the table 2.

1.5) Figure 4, any explanation between slopes of BAT and XRT tails?

R: The XRT light curve decays less steeply than BAT because of the evolution of the peak energy. While the peak energy crosses the XRT band, in the BAT band we observe the spectrum above the peak energy. This also explains why the XRT peak usually is slightly delayed with respect to the BAT peak. The different slopes in XRT and BAT bands as predicted by our model are showed in Fig. 1 of this document, where we used $\lambda = 1$ and an initial peak frequency $h\nu_p = 200$ keV.

A sentence about this is added in the caption of Fig. 1 in the paper.

1.6) Line 222, I believe alpha should be >3 instead of >2 .

R: We corrected it

1.7) Line 261-262: ”threshold N_0 is chosen case by case according to the brightness of the source.” This is vague. At least some specific numbers should be provided.

R: The specific numbers we used are reported few lines below: “A typical value of N_0^{WT} is around 1500-3000, while N_0^{PC} is around 500-1000”.

1.8) Line 318: I suggest replacing beta with a different symbol, as it is already used in Eq 2.

R: We replaced β with s .

name	SBPL			Band			Synchrotron		
	AC	HLE		AC	HLE		AC	HLE	
	χ^2/dof	χ^2/dof	Δ_{AIC}	χ^2/dof	χ^2/dof	Δ_{AIC}	χ^2/dof	χ^2/dof	Δ_{AIC}
090621A	1.13	2.94	8	1.56	2.86	6	1.26	3.19	9
100619A	0.35	12.1	69	0.23	11.6	66	1.59	12.8	67
110102A	0.50	29.8	145	0.10	28.6	141	1.11	30.2	145
140512A	10.3	47.2	193	9.6	46.1	190	13.2	19.6	43
161117A	0.80	17.5	132	0.62	16.3	124	3.18	19.6	133
170906A	1.91	16.7	148	2.81	16.2	135	3.58	17.9	145
180325A	0.43	20.9	80	0.74	20.1	76	0.57	23.8	91
190604B	0.30	12.8	61	0.40	12.5	59	0.37	13.6	65

Table 2: Comparison of best fit statistics between AC and HLE, adopting a SBPL, a Band function or a Synchrotron spectrum.

Figure 1: Flux density evolution computed at XRT and BAT frequencies

1.9) Line 431-432: I believed that "overall velocity and spectral distribution of mini-jets be the same for all the GRBs...to reproduce the $\alpha - F$ relation" is overstated. This has to be calculated, fitted and proved in more detail.

R: Following the relevant literature about this topic (e.g. [1],[2],[3],[4],[5]) we use the common approach to model the mini-jets, which consists in assuming an anisotropy of the intrinsic emission. In order to model such anisotropy we adopt an angular distribution of the emission in the form $P(\theta') \propto (\sin \theta')^n$, where n is the degree of anisotropy and θ' is the angle between the direction of the emitted photons and the local radial direction, as measured in the comoving frame. We evaluate the resulting HLE flux received by the observer as

$$F_\nu \propto P(\theta') \mathcal{D}^2(\theta) S \left(\frac{\nu}{P(\theta') \mathcal{D}(\theta) \nu'_c} \right)$$

where the dependence on time is implicit in θ . The resulting $\alpha - F$ relation for $n = 2$, $n = 5$ and $n = 10$ is showed in Fig. 2 of this document. The figure demonstrates that the larger the value of n , the more the predicted curves move away from the data. Since for $n = 0$ we are in the limit of standard HLE, which is already disfavoured by our study, we conclude that also mini-jets are not able to successfully reproduce the $\alpha - F$ relation.

The discussion above and the related figures are added in the Supplementary Information file.

2 Reviewer 2 (Remarks to the Author):

The paper "Unveiling the origin of steep decay in gamma-ray bursts" by Ronchini et al. provides a novel way of understanding the origin of the X-ray steep decay in GRBs. The paper is well written, concise and provides a comprehensive study on understanding the X-ray steep decay in GRBs, which is an important problem, since it is deeply connected to the nature of the prompt GRB emission process. There are some points that could be improved in the paper, but overall the paper is quite interesting, well written and certainly advances our study of these enigmatic phenomena.

There are two points that require some attention. These two points have already been addressed by the authors but could be emphasized/clarified.

2.1) The first one and more important is the fact that the relevant timescale obtained in their analysis is on the order of a second to a few seconds. This is quite large compared to the typical duration of GRB pulses, which last much less than a second. This should be emphasized more clearly in the paper as an inconsistency of their model.

R: The adiabatic cooling timescale we find reflects the decay timescale of the pulse tail. The fact that the analysed pulses show a longer decay timescale than typical prompt emission pulses could be due to the following reason. For the spectral analysis to be feasible, we had to choose only tails that are long enough (to be divided down into a sufficient number of temporal bins). Moreover, the prompt emission is usually interpreted as a superposition of several emission episodes: the steep decay observed in XRT is likely dominated by the tails with the slowest decay timescales. Since, in our model, the decay timescale is $\tau_{ad} = R/2c\Gamma^2$, this could indicate that the emission radius

Figure 2: Predicted $\alpha - F$ relation in case of mini-jets model, for three values of n , i.e. $n = 2$ (top left), $n = 5$ (top right) and $n = 10$ (bottom).

of the pulses that dominate the tail is systematically larger than that of pulses that dominate the prompt emission. If this is the case, a lower magnetic field is also expected, which goes well along with the long radiative timescale and slow (or marginally fast) cooling regime, in agreement with our results.

This discussion is added at the end of section “Adiabatic cooling”.

2.2) The second point is the mention toward the end of the Introduction of the $\alpha - F$ relation. Although the results seem to show that this relation is robust for their samples, it remains to be seen and studied if it applies to all bursts with available X-ray steep decay data.

R: In order to properly investigate the origin of spectral evolution in the steep decay, we need to select X-ray tails clearly connected to bright prompt emission events (ensured by the presence of a bright counterpart in the BAT band). Moreover, our approach requires a well-constrained peak flux, in order to test the spectral evolution against our model for the $\alpha - F$ relation. As discussed in the paper, in some cases it is possible to estimate the X-ray peak flux even if it was prior to the start of XRT light curve: this requires the occurrence of an isolated peak in the BAT band, which provides the peak time for a backwards extrapolation of the XRT light curve. As correctly noted by the reviewer, this limits the sample on which we can prove our results. Summarizing, the main selection criteria that reduce significantly the size of our samples are: 1) presence of a bright X-ray steep decay phase, not contaminated by secondary flares; 2) a well-defined peak of the tail, either in XRT or in BAT; 3) bright emission in the BAT band at the time of the peak. Cases lacking any of these requirements cannot be tested with our model.

In conclusion, we agree that a comprehensive analysis of all GRBs with steep decay would be of great interest for the community, but on the other hand with such approach a direct comparison with models would be infeasible or in any case largely biased by several unknowns, such as the peak flux or the initial spectral properties of the tail. The choice of restrictive selection criteria is unavoidable to perform a well targeted analysis of X-ray tails and to achieve robust conclusions about their origin.

In order to clarify how the conclusions depend on the selection criteria, we added the following sentence at the end of the section Extending the sample: “Thus our results are proved for X-ray tails firmly connected to prompt emission pulses.”

There are some other points that could be addressed, although some of these are suggestions:

2.3) Since the authors are using the peak in BAT, can the analysis of the authors extend to the BAT data as well and understand the decay seen in the BAT pulse considered in the paper? In this case, can the model explain the sharp peak in the BAT pulse and the broader peak in the X-ray data (see, e.g., Fig. 1 left panel)?

R: Yes, our analysis can be extended to include BAT data, but, as noted by the reviewer, the BAT flux has a much steeper drop with respect to XRT. Therefore, the shorter duration of the BAT tail introduces several difficulties in performing a time-resolved spectral analysis with a large enough S/N in each time bin. Nevertheless, we followed the suggestion and tried to analyse the case of 161117A, which shows one of the longest BAT tails in our sample. We divided the tail in 5 time bins and analysed the spectrum combining XRT+BAT data. For each bin, the spectrum is well fitted with a Band function. We first tried to leave all the parameters (i.e. the low- and

Figure 3: $E_0 - F$ relation for 161117A, compared with the expected evolution in case of adiabatic cooling. E_0 refers to the break energy of the Band function used to fit XRT+BAT data during the tail emission. The flux is computed in the BAT band 15-150 keV.

high-energy photon indices, the break energy E_0 , and the normalization) free to vary independently in all bins. In this case the fit parameters are poorly constrained and any comparison with our model is unfeasible. Therefore we tried to keep fixed in time both the low- and the high-energy photon indices, while leaving E_0 free to evolve, consistently with the assumption of a spectrum that shifts rigidly in frequency and normalization, as done in the paper. In this case, the goodness of fit is acceptable, the evolution of E_0 is well-constrained and the result can be compared with our theoretical expectations. The comparison is showed in Fig. 3 of this document, which is similar to the $\alpha - F$ relation, but where we replace α with E_0 . The dashed lines represent the evolution predicted by our model, for several values of λ . The agreement with the data is quite remarkable, confirming the solidity of our results. Unfortunately, applying this procedure to all the tails of our sample is not helpful for our purposes, since not all the BAT tails of our sample are long and bright enough to perform the same analysis.

2.4) The model yields a prediction for the peak energy as a function of time as it crosses the X-ray band. Does the time-resolved XRT analysis show this? This is probably hard to see due to the narrowness of the XRT band, but it is worth studying.

R: Unfortunately, the spectral shape is too smooth to appreciate its curvature within the XRT band. This has been verified by trying a fit of the XRT spectra with a broken power law. The result is that the fit is not able to converge and the parameters of the two power laws cannot be

constrained, given the quality of our data.

2.5) The inset of the right panel of Figure 2 refers to the 3 spectra, not only the blue spectra: The current figure makes the reader think that the inset only zooms-in into the blue line. This can be clarified.

R: The sketch has been modified and an explanatory sentence has been added in the caption.

2.6) The authors mention that the shell thickness choice is irrelevant since it does not change the result significantly, perhaps a figure addressing this could be included in the supplementary material.

R: We added the plot in the Supplementary Information file.

2.7) There is a minor typo in equation (18), (dR/dt) should be squared, however, equation (19) is correct.

R: We thank the reviewer and we corrected the typo.

3 Reviewer 3 (Remarks to the Author):

In this paper, the authors analyzed X-ray data in the steep decay phase, which has been interpreted as emission from a higher latitude. This paper shows a relation between the X-ray flux and photon-spectral index. The photon index evolves fast with decaying flux. This spectral evolution is significantly faster than expected in the high latitude emission model. This unexpected finding has great merit to be published. This might change the standard picture of GRB emission mechanism. However, the proposed model to explain their results in this paper has several unnatural points. I feel that the claim "evidence of adiabatic cooling" is too strong.

3.1) The revealed thing in this paper is the evolution of the photon index with the decaying flux. This behavior is consistent with the model shifting the spectral shape to lower energy and lower flux with the same functional shape. The unchanged spectral shape is consistent with the adiabatic cooling. However, the shift of the spectral peak energy or unchanged-but-shifted spectral shape itself is not verified. The adiabatic cooling model is not unique solution for the index evolution in this paper. Other types of non-trivial spectral-flux evolution can agree with the results in this paper.

R: In order to answer properly to this point, we carried out a deeper study of the cooling processes and their role in the modification of the spectrum. In particular, we started from the basic cooling equation, which in the generic form can be written as:

$$\frac{\partial N}{\partial t} = \dot{N}_{inj} - \frac{\partial}{\partial \gamma}(N\dot{\gamma})$$

where $N(\gamma) = dN_e/d\gamma$ (N_e being the number of emitting particles) and in the expression of $\dot{\gamma}$ we included both adiabatic and synchrotron cooling. In order to reproduce the effect of prompt-emission shut down, we assume an injection term $\dot{N}_{inj} \neq 0$ during the prompt phase $t < t_{inj}$, while $\dot{N}_{inj} = 0$ for $t > t_{inj}$. At any time the energy dependence of \dot{N}_{inj} is a power law, i.e. $N \propto \gamma^{-p}$,

Figure 4: $\alpha - F$ relation when the adiabatic and radiative time scales are comparable. The model is compared with the extended sample. The adopted spectral shape is a SBPL with a cut-off close to the F_ν peak.

for $\gamma > \gamma_m$. We numerically solved the cooling equation, including the time-dependence of the magnetic field that we introduced in the paper, i.e.

$$B(R) = B_0 \left(\frac{R}{R_0} \right)^{-\lambda}$$

For $t > t_{inj}$, namely in the post-prompt phase corresponding to the tail emission, the resulting scenarios can be summarised as follows:

1. If the particles synchrotron cooling timescale is much larger than the dynamical time scale, adiabatic cooling dominates the evolution of the distribution. In this limit, the particle distribution is shifted rigidly to lower energies, since the energy of each particle evolves in the same way, i.e. $\gamma \propto t^{-2/3}$ (in case of constant shell thickness). This means that the slope p of the injection spectrum is not affected by adiabatic losses.
2. If the synchrotron cooling timescale is much smaller than the dynamical time scale, all the particles radiate almost instantaneously. This would bring us back to the ideal case of an abrupt shutdown of the emitting surface, which we used to model the HLE and which is ruled out in our conclusions. In case of emission dominated by the last emitting surface, the entire spectral evolution is given by the shift of the spectrum in the observed frame due to the gradual drop of the Doppler factor as a function of the angle.

Figure 5: Left: Evolution in time of the synchrotron spectral shape for a decaying particle injection (in this case we adopted $y=3$) and a constant magnetic field. The time goes from $t = t_{inj}$ up to $t = 10t_{inj}$. Right: corresponding $\alpha - F$ relation imposing that the observing band is below the initial spectral peak, which ensures that the initial photon index is $\sim 2/3$.

3. There is an intermediate case, when the two time scales are comparable. At any given time, synchrotron completely erases all particles above the cooling energy γ_c because no additional particles are injected, introducing an exponential cut-off which moves gradually towards lower energies. Hence the resulting particle distribution would be:

$$N \propto \gamma^{-p} e^{-\gamma/\gamma_c}$$

The presence of a cut-off around $\gamma_c \sim \gamma_m$ can explain the large values of photon indices reached in the extended sample. The corresponding $\alpha - F$ relation in case of adiabatic cooling and mild synchrotron cooling is showed in Fig. 4 of this document, which is added in the paper.

The agreement of the data with case 1 or at most with case 3 confirms our conclusion that the particles are in a slow or moderately fast cooling regime and that adiabatic cooling is an important ingredient of the spectral evolution.

As suggested later by the reviewer in point 3.3, there is the possibility that a gradual decrease of the injection term can introduce a softening of the spectrum. We tested also this possibility taking $\dot{N}_{inj} = \text{const}$ during the prompt phase $t < t_{inj}$ and $\dot{N}_{inj} \propto (t/t_{inj})^{-y}$ for $t > t_{inj}$, with $y > 0$. In this case, we excluded adiabatic cooling in order to see if a decaying particle injection alone can reproduce the observed softening. We first considered the case of constant magnetic field and we verified that a decaying \dot{N}_{inj} gives a gradual depletion of particles above the cooling energy γ_c . The depletion is more severe for larger values of y and in the limit of $y \rightarrow \infty$ we recover the cutoff discussed in point 3 above. The particle depletion creates a hump in $N(\gamma)$ at low energies and the corresponding evolution of the synchrotron shape is showed in the left panel of Fig. 5 of this document. Imposing that at the beginning the F_ν peak is just above the observing band, the resulting spectral evolution would be an initial softening followed by a hardening, as showed in right panel of Fig. 5 of this document.

In the case of a decay of both the magnetic field and \dot{N}_{inj} , the corresponding effects on the spectral shape tend to compensate each other, as found also by Panaitescu 2019 [7]. As shown in Fig. 6 of this document, the general result is that, increasing y , the effect of decaying \dot{N}_{inj} becomes dominant

Figure 6: Temporal evolution of the particle distribution for a decay of both \dot{N}_{inj} and B. The adopted parameters are $y = 2$ and $\lambda = 2$ for the left panel, $y = 4$ and $\lambda = 1$ for the right panel. The evolution is followed from $t = t_{inj}$ (the orange line, which is the standard γ^{-2} cooling branch of the distribution) up to $t = 10t_{inj}$.

and we recover results similar of those in Fig. 5, while increasing λ the effects of a decaying B become dominant and the cooling tail tends to harden, instead of softening. If the two effects are comparable, the slope of the cooling tail of the particle distribution is barely modified, meaning that the corresponding synchrotron spectrum has a photon index that goes from $2/3$ to $3/2$ crossing the cooling frequency ν_c , which is not consistent with our data.

On the basis of these tests, we conclude that intrinsic modifications of the spectral shape can hardly give an agreement with data comparable to the case of a rigidly shifting spectrum.

The discussion reported above is added in the Supplementary Information file, along with Figures 5 and 6.

3.2) As shortly mentioned in 3. Conclusions, if adiabatic cooling is the dominant cooling process, the radiative cooling timescale should be longer than the dynamical timescale. The electrons with average Lorentz factor $\langle \gamma \rangle$ are roughly responsible for the spectral peak energy $\sim \text{MeV}$. Achieving this peak energy and long radiative cooling timescale are compatible? Such discussion is lacking. Or the authors strongly support proton synchrotron model? If so, the authors should emphasize proton synchrotron rather than adiabatic cooling. In this case, is the jet energy dominated by protons or magnetic field? How much is the energy conversion efficiency to photons?

R: As clarified in point 3.1, our results suggest that particles are in slow cooling but possibly also in marginally fast cooling (see again Fig. 5 of this document). Therefore, it is not guaranteed that radiative cooling is strongly subdominant with respect to adiabatic cooling. Nevertheless, even if the cooling is marginally fast, an extremely small magnetic field would be required for electrons to be responsible for the emission (as pointed out in Ghisellini et al. 2020 [6]). Protons alleviate this problem and therefore are supported by our results. We follow the advice of the reviewer and we now stress this implication more in the text.

The adiabatic timescale that we find (around 1-10 s) for our pulses implies an emission radius of $\sim 10^{15}$ cm (taking $\Gamma \sim 100$), which is ~ 100 times the typical prompt emission radius in case of internal shocks. This could indicate that the steep decay observed by XRT is dominated by the

tails of the pulses with the longer decay time scales, namely those emitted at larger radii. Therefore, if the magnetic field decays with radius, the radiative cooling timescale increases consequently, explaining why adiabatic cooling starts to be relevant.

Regarding the jet energy composition and the conversion efficiency, our analysis does not give us enough elements to come to a conclusive answer on these issues.

3.3) The HLE model and adiabatic cooling model are compared. However, while the cooling process for the adiabatic cooling model is calculated, a sudden shutdown of the emission without spectral softening is assumed in the HLE model. This may be not even treatment. Even in the HLE model, the spectral shape evolves in the decaying phase with particle cooling and decaying particle injection, probably showing softening. The late time such emission from high latitude may contribute to the emission.

R: As mentioned before, the assumption of an abrupt shutdown of the emission is valid only in case of efficient cooling of particles on time scales much shorter than the dynamical time scales. This is what we mean by modelling pure HLE, namely the emission power from the jet surface drops so fast that after the shutdown the flux and spectral evolution are dominated by photons departed from the last emitting surface. Hence, if our HLE model does not reproduce the observed $\alpha - F$ relation, this implies that an efficient cooling of particles cannot account for the spectral evolution in the X-ray tails, rather than being a proof of incompatibility with data of HLE itself. If instead particles do not cool efficiently, the assumption of abrupt shutdown is no more valid and an integration along the EATS is necessary to account for contributions from different angles, as consistently done in the paper. The effect of combined radiative cooling and decaying particle injection is discussed in point 3.1.

In order to clarify this point in the paper, now we have specified several times that the treatment of an abrupt shutdown relies on the assumption of efficiently cooled particles. This concept is also remarked in the last section in Supplementary Notes.

3.4) I can not understand the phrase in abstract "evidence of ... efficient energy exchange between the emitting particles in the relativistic outflows of GRBs." Adiabatic means no energy exchange. Particle energy is not efficiently released as photons. So most of the energy may remain in the jet not being released.

R: We convene that the sentence may be misleading and therefore we removed it.

3.5) Re-acceleration/slow heating of electrons is rejected in this paper. This statement is also based on the sudden shutdown assumption. Even in such models, the spectra may show gradual softening with decaying re-acceleration or heating rate (e.g. Asano & Terasawa 2105). This may show a similar behavior to the adiabatic cooling model.

R: The original, simplest slow-heating scenario discussed in Asano & Terasawa 2009 assumes the magnetic field suddenly decays at some distance from the shock, where its generation by turbulence becomes ineffective. This is a viable alternative to explain the presence of a $2/3$ low-energy photon index in the spectra, as particles stop emitting before being able to cool completely (as long as $t_{\text{dec}} \sim t_{\text{cool}}$, as discussed in that paper). On the other hand, in this scenario, as soon as the shock crosses the shell (and therefore particle injection stops), particle acceleration is halted along

with the generation of magnetic field (as both rely on the presence of shock-generated turbulence), leading to an abrupt switch-off of the emission. This leads again to HLE being the dominant effect in determining the tail flux and spectral evolution, which is clearly disfavoured by our analysis. A slower decay of the magnetic field after the shock crossing, such as that discussed in the second part of the Asano & Terasawa 2009 paper (and in more detail in the Asano & Terasawa 2015 paper), along with a decaying particle acceleration, could be compatible with our results, but we still need adiabatic cooling (which is anyway unavoidable) to play the leading role in the spectral evolution, that is, we still need the radiative cooling time scale t_{cool} to be $\gtrsim t_{\text{ad}}$ during this phase (see point 3.1). We therefore conclude that, while the slow-heating scenario is not *per se* rejected by our results, it cannot be invoked as the main mechanism behind the α - F relation.

The sentence “However, our results disfavor models with re-acceleration/slow heating of electrons, since when electrons are left unenergized we would again observe HLE-dominated X-ray tails” in the Conclusions is removed, while the discussion reported above is added in the Supplementary Notes.

References

- [1] Narayan, R. & Kumar, P. 2009, A turbulent model of gamma-ray burst variability, *mnras*, 394, L117. doi:10.1111/j.1745-3933.2009.00624.x
- [2] Barniol Duran, R., Leng, M., & Giannios, D. 2016, An anisotropic minijets model for the GRB prompt emission , *mnras*, 455, L6. doi:10.1093/mnrasl/slv140
- [3] Yamazaki, R., Toma, K., Ioka, K., et al. 2006, Tail emission of prompt gamma-ray burst jets, *mnras*, 369, 311. doi:10.1111/j.1365-2966.2006.10290.x
- [4] Beloborodov, A. M., Daigne, F., Mochkovitch, R., et al. 2011, Is gamma-ray burst afterglow emission intrinsically anisotropic? , *mnras*, 410, 2422. doi:10.1111/j.1365-2966.2010.17616.x
- [5] Geng, J.-J., Huang, Y.-F., & Dai, Z.-G. 2017, Steep Decay of GRB X-Ray Flares: The Results of Anisotropic Synchrotron Radiation, *apjl*, 841, L15. doi:10.3847/2041-8213/aa725a
- [6] Ghisellini, G., and 8 colleagues 2020. Proton-synchrotron as the radiation mechanism of the prompt emission of gamma-ray bursts?. *Astronomy and Astrophysics* 636, A82.
- [7] Panaitescu, A. 2019. Adiabatic and Radiative Cooling of Relativistic Electrons Applied to Synchrotron Spectra and Light Curves of Gamma-Ray Burst Pulses. *The Astrophysical Journal*, 886, 106

REVIEWER COMMENTS

Reviewer #1 (Remarks to the Author):

I appreciate the authors' effort in improving the paper. However, given the current presentations, unfortunately, I am still not convinced by their conclusion. Specifically, from the observer's perspective, what we have for each burst is the energy flux and spectral indices. Any successful model should reproduce such observed data as precisely as possible, at a statistically-accepted level. I did not see this in the draft. Although the authors tried to do so in Figure 3, the presentations are not convincing. The model (e.g., with $\lambda=0.4$) appears to follow the burst sample's overall trends, but it doesn't guarantee a good fit. Even though the authors tried individual burst fit, the χ^2/dof values listed in Table 1 seem rather diverse. I believe if the authors calculate the p-value using such χ^2 and dof, it will yield a lot of overfitting or unsuccessful fit.

I understood the proposed $\alpha-F/F_{\text{max}}$ is impressive, and the trend seems consistent with the data. However, consistency does not mean a match. Thus, I strongly recommend the authors make the *same* plot as Figure 2 of Zhang, 2012, ApJ, 756, 190 for each burst. Please compare the model predicted flux and photon index values directly to the observed ones and list the fitting results. So the readers can have clear evidence if data support the model. (Please note, a Monte Carlo fitting technique such as ememc may be required here in order to fully explore the parameter space).

Reviewer #2 (Remarks to the Author):

The authors of "Unveiling the origin of steep decay in gamma-ray bursts" have provided a very comprehensive response to all queries. The paper is, in my opinion, ready to be accepted for publication.

A minor suggestion is to include references to Tables 1 and 2 in Section 2 when discussing the results from these tables. A bit more explanation in the Table 1 caption (explaining that λ and ν_p

are fixed to the fitted values in order to fit the light curve and obtain τ_{ad}) and in the Table 2 caption (explaining what Δ_{AIC} is - Akaike Information Criterion - and that AC is significantly favored with respect to HLE) could be included. Also, Figures could be numbered in the order they are mentioned in the main text.

Reviewer #3 (Remarks to the Author):

Authors satisfactorily dealt with most of the points I asked.

In addition, they performed additional calculations including several evolutionary effects of the parameters. This may help readers understand the role of adiabatic cooling.

If the authors prefer the proton synchrotron model, this should be mentioned in the abstract.

One point I'd like to comment.

Of course, adiabatic cooling always exists.

I also have understood that the proposed model with adiabatic cooling can explain the observed data.

I agree that the paper focuses on this adiabatic cooling model.

But I still think that a model where the adiabatic cooling is subdominant effect can be possible.

Not only the injection rate but also the minimum Lorentz factor, spectral index, and/or heating rate can evolve with time. We have so many options of the evolution of the parameters, which were not comprehensively tested.

I do not require additional calculations.

But I think we cannot conclude "Adiabatic cooling results to be a dominant component" as written in the abstract.

This is one of possible scenarios.

Point by point reply to reviewers

1 Reviewer 1 (Remarks to the Author):

I appreciate the authors' effort in improving the paper. However, given the current presentations, unfortunately, I am still not convinced by their conclusion. Specifically, from the observer's perspective, what we have for each burst is the energy flux and spectral indices. Any successful model should reproduce such observed data as precisely as possible, at a statistically-accepted level. I did not see this in the draft. Although the authors tried to do so in Figure 3, the presentations are not convincing. The model (e.g., with $\lambda=0.4$) appears to follow the burst sample's overall trends, but it doesn't guarantee a good fit. Even though the authors tried individual burst fit, the χ^2/dof values listed in Table 1 seem rather diverse. I believe if the authors calculate the p-value using such χ^2 and dof, it will yield a lot of overfitting or unsuccessful fit.

I understood the proposed α - F/F_{max} is impressive, and the trend seems consistent with the data. However, consistency does not mean a match. Thus, I strongly recommend the authors make the *same* plot as Figure 2 of Zhang, 2012, ApJ, 756, 190 for each burst. Please compare the model predicted flux and photon index values directly to the observed ones and list the fitting results. So the readers can have clear evidence if data support the model. (Please note, a Monte Carlo fitting technique such as emcee may be required here in order to fully explore the parameter space).

A:

We thank the referee for the comment, which helped to improve the analysis and to further strengthen our results. We repeated the comparison of our model (adiabatic cooling + HLE) with data directly representing the flux as a function of time and photon index as a function of time. In order to fully explore the parameter space of the model, we performed a MCMC, using the emcee algorithm [1]. The setup of our analysis is described in the following points:

- The model contains as free parameters E_p , λ and $\tau_{ad} = R/2c\Gamma^2$, which are the peak energy at the beginning of the steep decay, the decay index of the magnetic field, and the adiabatic time scale. The inclusion of Γ as free parameter returns a flat posterior distribution, indicating that the model is insensitive to it. Therefore we performed the analysis fixing $\Gamma = 100$.
- The MCMC is performed jointly for flux and photon index evolution. The adopted likelihood is:

$$\log(\mathcal{L}) = -\frac{1}{2} \sum_n \left[\frac{(\phi_n - \bar{\phi}(t_n))^2}{s_{\phi,n}^2} + \ln(2\pi s_{\phi,n}^2) \right] - \frac{1}{2} \sum_n \left[\frac{(\alpha_n - \bar{\alpha}(t_n))^2}{s_{\alpha,n}^2} + \ln(2\pi s_{\alpha,n}^2) \right]$$

where $\phi = F/F_{\text{max}}$, α is the photon index and with $\bar{\phi}$, $\bar{\alpha}$ we indicate the value predicted by

the model at each time t_n . Moreover,

$$s_{\phi,n}^2 = \sigma_{\phi,n}^2 + f_{\phi} \cdot \bar{\phi}(t_n), \quad s_{\alpha,n}^2 = \sigma_{\alpha,n}^2 + f_{\alpha} \cdot \bar{\alpha}(t_n)$$

where σ_{ϕ} and σ_{α} are the errors, while f_{ϕ} and f_{α} are introduced to take into account possible underestimation of the errors. Since keeping $f_{\alpha} \neq 0$ leads to a posterior distribution of f_{α} peaked around $\sim 10^{-7}$, the parameter estimation was performed fixing $f_{\alpha} = 0$. Instead, we keep $f_{\phi} \neq 0$ taking into account that the error on the flux resulting from the fitting of time-averaged spectrum may not represent the true flux error over the time bin.

- The MCMC runs until the number of steps exceeds 100 times the autocorrelation time (its maximum) and the averaged autocorrelation time (over 100 steps) becomes constant within 1% accuracy. The burn-in is chosen as twice of autocorrelation time. As an example, we show in Fig. 1 the evolution of the autocorrelation time as a function of the steps.

The resulting parameter estimation is summarized in Tab. 1. The uncertainties are reported based on the 16th, 50th, and 84th percentiles of the samples in the marginalized distributions (1σ level of confidence). An example of corner plot obtained via MCMC is shown in Fig. 2. In Fig. 3 - 4, we show for each burst the observed temporal evolution of photon index and normalized flux in comparison with the curves produced with 500 random draws from the posterior sample set of the MCMC. Such direct comparison shows an incontrovertible match of the model with the observations.

We performed an analogous MCMC analysis adopting the model of HLE from an instantaneous emission. However, the algorithm is unable to converge, demonstrating that the model cannot successfully match with the observations. The only way to obtain converged chains by this model is to admit extreme and unrealistic values of f_{ϕ} , of the order $10^4 - 10^8$. The only exception is GRB 090621, which can be fitted by HLE alone. This is the only case where it is meaningful to compute the Bayes factor between HLE and adiabatic cooling, which results to be ~ 200 . Thus we prove that the adiabatic cooling model is strongly preferred for all the analyzed cases.

The Tab. 1 is included in the main text and we replace the previous table where we listed the results coming from the fitting of the $\alpha - F$ relation. The illustrative corner plot and the simultaneous temporal evolution of flux and photon index, compared to our model, are inserted in the Supplementary material.

This further test gives a firm statistical evidence which confirms and strengthen our results.

2 Reviewer 2 (Remarks to the Author):

The authors of "Unveiling the origin of steep decay in gamma-ray bursts" have provided a very comprehensive response to all queries. The paper is, in my opinion, ready to be accepted for publication.

A minor suggestion is to include references to Tables 1 and 2 in Section 2 when discussing the results from these tables. A bit more explanation in the Table 1 caption (explaining that λ and nu_p are fixed to the fitted values in order to fit the light curve and obtain τ_{ad}) and in the Table 2 caption (explaining what Delta AIC is - Akaike Information Criterion - and that AC is

significantly favored with respect to HLE) could be included. Also, Figures could be numbered in the order they are mentioned in the main text.

A:

We thank the referee for the suggestions, which we have implemented in the draft.

3 Reviewer 3 (Remarks to the Author):

Authors satisfactorily dealt with most of the points I asked. In addition, they performed additional calculations including several evolutionary effects of the parameters. This may help readers understand the role of adiabatic cooling.

If the authors prefer the proton synchrotron model, this should be mentioned in the abstract.

One point I'd like to comment. Of course, adiabatic cooling always exists. I also have understood that the proposed model with adiabatic cooling can explain the observed data. I agree that the paper focuses on this adiabatic cooling model. But I still think that a model where the adiabatic cooling is subdominant effect can be possible. Not only the injection rate but also the minimum Lorentz factor, spectral index, and/or heating rate can evolve with time. We have so many options of the evolution of the parameters, which were not comprehensively tested. I do not require additional calculations. But I think we cannot conclude "Adiabatic cooling results to be a dominant component" as written in the abstract. This is one of possible scenarios.

A:

We agree with the referee that alternative scenarios cannot be definitively excluded given the complexity of the system and the plethora of parameters at play. According to the request of the referee, we modified the formulation of our conclusions and the abstract. However, based on the extensive tests we performed and the various alternative scenarios we explored, we are confident that the adiabatic cooling process remains the most plausible and natural scenario able to explain the observations.

References

- [1] Foreman-Mackey, D., Hogg, D. W., Lang, D., et al. 2013, *pas*, 125, 306. doi:10.1086/670067
-

GRB	E_{peak}	λ	τ_{ad}
	keV		s
090621A	18_{-2}^{+3}	$2.11_{-0.54}^{+0.56}$	$24.4_{-3.0}^{+4.7}$
100619A	> 129	$0.47_{-0.07}^{+0.11}$	$0.3_{-0.2}^{+1.0}$
110102A	46_{-9}^{+15}	$0.61_{-0.10}^{+0.10}$	$5.8_{-1.1}^{+1.9}$
140512A	> 323	$0.48_{-0.03}^{+0.04}$	$0.9_{-0.4}^{+0.9}$
161117A	80_{-21}^{+55}	$0.69_{-0.10}^{+0.10}$	$6.2_{-2.3}^{+2.0}$
170906A	135_{-53}^{+204}	$0.66_{-0.09}^{+0.10}$	$3.0_{-1.5}^{+1.6}$
180325A	> 122	$0.39_{-0.05}^{+0.06}$	$0.8_{-0.5}^{+1.3}$
190604B	54_{-20}^{+227}	$0.45_{-0.15}^{+0.25}$	$3.5_{-2.8}^{+2.6}$

Table 1: Results of the parameter estimation via MCMC, adopting the adiabatic cooling model. The confidence intervals and the lower limits represent the 16th, 50th, and 84th percentiles of the samples in the marginalized distributions (i.e. 1σ level of confidence).

Figure 1: An example of autocorrelation time-MCMC steps plot (GRB 161117A). The line shows the $100\times$ autocorrelation time.

Figure 2: An example of corner plot from the MCMC (GRB 161117A).

Figure 3: Joint temporal evolution of normalized flux and photon index (blue points), compared to the best fit curve of the adiabatic cooling model (black line). The orange lines are curves produced with 500 random draws from the posterior sample set of the MCMC.

Figure 4: -Continued-

REVIEWERS' COMMENTS

Reviewer #1 (Remarks to the Author):

The authors have fully answered my questions with substantial evidence. I am glad to recommend it for publication in Nature Communications.

Reviewer #2 (Remarks to the Author):

I thank the authors for including my minor suggestions in this revision of the draft. The paper is, in my opinion, ready for publication.

Reviewer #3 (Remarks to the Author):

No further comment.